# Pruning Spurious Subgraphs for Graph Out-of-Distribtuion Generalization

## Abstract

Graph Neural Networks (GNNs) often encounter significant performance degradation under distribution shifts between training and test data, hindering their applicability in real-world scenarios. Recent studies have proposed various methods to address the out-of-distribution (OOD) generalization challenge, with many methods in the graph domain focusing on directly identifying an invariant subgraph that is predictive of the target label. However, we argue that identifying the edges from the invariant subgraph directly is challenging and error-prone, especially when some spurious edges exhibit strong correlations with the targets. In this paper, we propose `PrunE`, the first pruning-based graph OOD method that eliminates spurious edges to improve OOD generalizability. By pruning spurious edges, `PrunE` retains the invariant subgraph more comprehensively, which is critical for OOD generalization. Specifically, `PrunE` employs two regularization terms to prune spurious edges: 1) *graph size constraint* to exclude uninformative spurious edges, and 2) $\epsilon$-*probability alignment* to further suppress the occurrence of spurious edges. Through theoretical analysis and extensive experiments, we show that `PrunE` achieves superior OOD performance and outperforms previous state-of-the-art methods significantly. Codes will be available upon acceptance.

## 1. Introduction

Graph Neural Networks (GNNs) (Kipf & Welling, 2017; Xu et al., 2018; Veličković et al., 2017) often encounter significant performance degradation under distribution shifts between training and test data, hindering their applicability in real-world scenarios (Hu et al., 2020; Huang et al., 2021; Koh et al., 2021). To address the out-of-distribution (OOD)

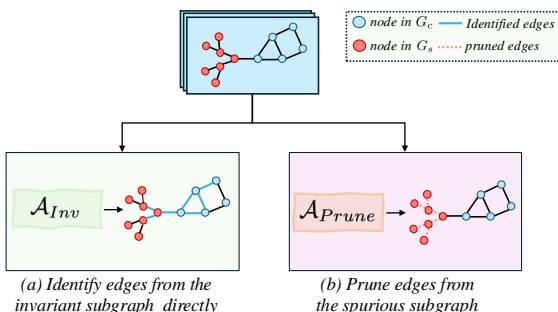

*Figure 1.* Illustration of two learning paradigms for graph-specific OOD methods. Previous methods seek to identify edges from the invariant subgraph directly, while our approach prunes edges from the spurious subgraph, which is more effective at preserving the invariant substructure.

generalization challenge, recent studies propose to utilize the causally invariant mechanism to learn invariant features that remain stable across different environments (Peters et al., 2016; Arjovsky et al., 2020; Ahuja et al., 2021; Jin et al., 2020; Krueger et al., 2021; Creager et al., 2021). In graph domain, various methods have been proposed to address the OOD generalization problem (Wu et al., 2022b; Li et al., 2022b; Chen et al., 2022; Liu et al., 2022; Sui et al., 2023; Gui et al., 2023; Yao et al., 2024), Most OOD methods, both in the general domain and the graph domain, aim to learn invariant features directly. To achieve this, many graph-specific OOD methods utilize a subgraph selector to model independent edge probabilities to directly identify invariant subgraphs that remain stable across different training environments (Chen et al., 2022; Miao et al., 2022; Wu et al., 2022b; Sui et al., 2023). However, we argue that directly identifying invariant subgraphs can be challenging and error-prone, particularly when spurious edges exhibit strong correlations with target labels. In such scenarios, certain edges in the invariant subgraph $G_c$ may be misclassified (i.e., assigned low predicted probabilities), leading to *partial* preservation of the invariant substructure and thereby degrading OOD generalization performance. In contrast, while a subset of spurious edges may correlate strongly with targets, the majority of spurious edges are relatively uninformative and easier to identify due to their weak correlations with labels. Consequently, pruning these less informative edges is more likely to preserve the invariant substructure effectively, although some spurious edges may still persist.

[1]Anonymous Institution, Anonymous City, Anonymous Region, Anonymous Country. Correspondence to: Anonymous Author <anon.email@domain.com>.

Preliminary work. Under review by the International Conference on Machine Learning (ICML). Do not distribute.

In this work, we raise the following research question:

*Can we prune spurious edges instead of directly identifying invariant edges to enhance OOD generalization ability?*

To address this question, we propose the first *pruning-based* OOD method. Unlike most existing graph OOD methods that aim to directly identify edges in the invariant subgraph, our method focuses on pruning spurious edges to achieve OOD generalization (Figure 1). We first begin with a case study to investigate the differences between our method and previous ones that directly identify invariant subgraphs, in terms of how the induced subgraph selector estimates edges from the invariant subgraph $G_c$ and spurious subgraph $G_s$. Specifically, we observe that previous methods tend to misclassify some edges in $G_c$ as unimportant edges with low probabilities, while assigning high probabilities to certain edges in $G_s$. As a result, *the invariant substructure in the graph is not preserved.* In contrast, our pruning-based method preserves the invariant subgraph more effectively (i.e., estimating the edges in $G_c$ with high probabilities), although a small number of spurious edges may still remain due to the strong correlation with the targets. However, by preserving the invariant substructure more effectively, our method PrunE (**Prun**ing spurious **E**dges for OOD generalization) achieves enhanced OOD performance compared to previous approaches that directly identify invariant subgraphs.

The core insight behind PrunE is that Empirical Risk Minimization (ERM) (Vapnik, 1995) tends to capture all "useful" features that are correlated with the targets (Kirichenko et al., 2023). In our context, ERM pushes the subgraph selector to preserve substructures that are more informative for prediction. By forcing uninformative edges to be excluded, $G_c$ is preserved due to its strong correlation with the targets and the inherent inductive bias of ERM. To prune spurious edges, our proposed OOD objective consists of two terms that act on the subgraph selector, without adding additional OOD objective: 1) *graph size constraint.* This constraint limits the total edge weights derived from the subgraph selector to $\eta|G|$ for a graph $G$, where $\eta < 1$, thereby excluding some uninformative edges. 2) $\epsilon$-*probability alignment.* This term aligns the probabilities of the lowest $K\%$ edges to be close to zero, further suppressing the occurence of uninformative edges. Through theoretical analysis and extensive empirical validation, we demonstrate that PrunE significantly outperforms existing methods in OOD generalization, establishing state-of-the-art performance across various benchmarks. Our contributions are summarized as follows:

- **Novel framework.** We propose a *pruning-based* graph OOD method PrunE, which introduces a novel paradigm focusing on removing spurious edges rather than directly identifying edges in $G_c$. By pruning spurious edges, PrunE preserves more edges in $G_c$ than

previous methods, thereby improving its OOD generalization performance.

- **Theoretical guarantee.** We provide theoretical analyses, demonstrating that: 1) The proposed graph size constraint provably enhances OOD generalization ability by reducing the size of $G_s$; 2) The proposed learning objective (Eqn. 5) provably identifies the invariant subgraph by pruning spurious edges.

- **Strong empirical performance.** We conduct experiments on both synthetic datasets and real-world datasets, compare against 15 baselines, PrunE outperforms the second-best method by up to 24.19%, highlighting the superior OOD generalization ability.

## 2. Preliminary

**Notation.** Throughout this work, an undirected graph $G$ with $n$ nodes and $m$ edges is denoted by $G := \{\mathcal{V}, \mathcal{E}\}$, where $\mathcal{V}$ is the node set and $\mathcal{E}$ denotes the edge set. $G$ is also represented by the adjacency matrix $\mathbf{A} \in \mathbb{R}^{n \times n}$ and node feature matrix $\mathbf{X} \in \mathbb{R}^{n \times D}$ with $D$ feature dimensions. We use $G_c$ and $G_s$ to denote invariant subgraph and spurious subgraph. $\widehat{G}_c$ and $\widehat{G}_s$ denote the estimated invariant and spurious subgraph. $t : \mathbb{R}^{n \times n} \times \mathbb{R}^{n \times D} \to \mathbb{R}^{n \times n}$ refers to a learnable subgraph selector that models each independent edge probability, $\widetilde{G} \sim t(G)$ represents $\widetilde{G}$ is sampled from $t(G)$. We use $\mathbf{w}$ to denote a vector, and $\mathbf{W}$ to denote a matrix respectively. Finally, a random variable is denoted as $W$, a set is denoted using $\mathcal{W}$. A more complete set of notations is presented in Appendix A.

**OOD Generalization.** We consider the problem of graph classification under various forms of distribution shifts in hidden environments. Given a set of graph datasets $\mathcal{G} = \{G^e\}_{e \in \mathcal{E}_{\text{tr}}}$, a GNN model $f = \rho \circ h$, comprises an encoder $h \colon \mathbb{R}^{n \times n} \times \mathbb{R}^{n \times D} \to \mathbb{R}^F$ that learns a representation $\mathbf{h}_G$ for each graph $G$, followed by a downstream classifier $\rho \colon \mathbb{R}^F \to \mathbb{Y}$ to predict the label $\widehat{Y}_G = \rho(\mathbf{h}_G)$. In addition, a subgraph selector $t(\cdot)$ is employed to generate a graph with structural modifications. The objective of OOD generalization in our work is to learn an optimal composite function $f \circ t$ that encodes stable features by regularizing $t(\cdot)$ to prune spurious edges while preserving the edges in $G_c$.

**Assumption 2.1.** Given a graph $G \in \mathcal{G}$, there exists a stable subgraph $G_c$ for every class label $Y \in \mathcal{Y}$, satisfying: a) $\forall e, e' \in \mathcal{E}_{tr}, P^e(Y \mid G_c) = P^{e'}(Y \mid G_c)$; b) The target $Y$ can be expressed as $Y = f^*(G_c) + \epsilon$, where $\epsilon \perp\!\!\!\perp G$ represents random noise.

Assumption 2.1 posits the existence of a subgraph $G_c$ that remains stable across different environments and causally determines the target $Y$, thus is strongly correlated with the target labels. Our goal in this work is to identify edges in $G_c$ by excluding spurious edges to achieve OOD generalization.

## 3. Should We Identify Invariant Subgraphs or Prune Spurious Subgraphs?

In this section, we conduct a case study to explore the differences between previous graph OOD methods and our proposed approach in the estimated edge probabilities. Through experiments, we observe that our pruning-based method is more effective at preserving $G_c$ compared to previous methods that aim to directly identify $G_c$, thereby facilitating better OOD generalization performance for our approach. Next we detail the experimental setup and observations.

**Datasets.** We use GOODMotif (Gui et al., 2022) dataset with *base* split for the case study. More details of this dataset can be found in Appendix G.

**Experiment Setup.** We use GSAT (Miao et al., 2022), CIGA (Chen et al., 2022), and AIA (Sui et al., 2023) as baseline methods representing three different lines of work for comparison, all of which utilize a subgraph selector to directly identify $G_c$ for OOD generalization. After training and hyperparameter tuning, we obtain a model and a well-trained subgraph selector for each method. We evaluate the test performance on the Motif-base dataset and calculate the average number of edges in $G_c$ and $G_s$ among the top-$K$ predicted edges for all methods. Here, we set $K = 1.5|G_c|$. For our method, we also present the statistics under different values of $K$.

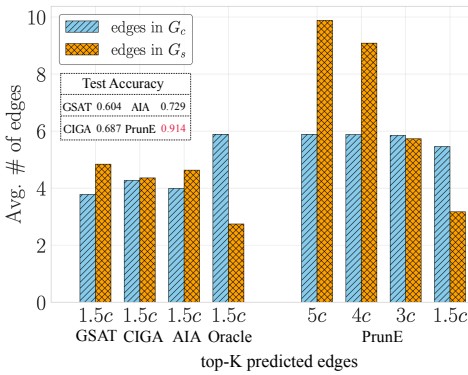

*Figure 2.* Illustration of the average number of edges from $G_c$ and $G_s$ included in the top-$K$ predicted edges, where $c$ denotes $|G_c|$.

**Observations.** From Figure 2, we observe that: (1) `PrunE` outperforms all the baselines by a significant margin, demonstrating superior OOD generalization ability; (2) When $K = 1.5|G_c|$, our method preserves more edges from $G_c$ compared to other methods. Moreover, as $K$ transitions from $5|G_c|$ to $1.5|G_c|$, the average number of edges in $G_c$ remains nearly constant, while the number of edges from $G_s$ decreases significantly. This indicates that most edges from $G_c$ have predicted probabilities greater than those from $G_s$; (3) When compared with the oracle, the average number of edges in $G_c$ under our method is still slightly lower than the oracle value, suggesting that a small number of spurious

edges are estimated with high probability.

In conclusion, compared to directly identifying invariant edges (i.e., edges in $G_c$), pruning spurious edges preserves more edges in $G_c$, even if some spurious edges remain challenging to eliminate. However, the OOD performance can be substantially improved by retaining the invariant subgraphs, which explains why our pruning-based method outperforms previous approaches. We also provide a detailed discussion in Appendix F on why traditional graph OOD methods tend to assign low probabilities to edges in $G_c$, and how our pruning-based approach avoids this pitfall. Next, we detail the design of our pruning-based method.

## 4. The Proposed Method

In this section, we present our pruning-based method `PrunE` , which directly regularizes the subgraph selector without requiring any additional OOD regularization. The pseudocode is shown in Algorithm 1.

**Subgraph selector.** Following previous studies (Ying et al., 2019; Luo et al., 2020; Wu et al., 2022b), we model each edge $e_{ij} \sim Bernoulli(p_{ij})$ independently which is parameterized by $p_{ij}$. The probability of the graph $G$ is factorized over all the edges, i.e., $P(G) = \prod_{e_{ij} \in \mathcal{E}} p_{ij}$. To parameterize $\mathcal{T}_\theta$, we employ a GNN model to derive the node representation for each node $v$, followed by an MLP to obtain the logits $w_{ij}$ as following:

$$
\begin{aligned}
\mathbf{h}_v &= \text{GNN}(v \mid G), \ v \in \mathcal{V}, \\
w_{ij} &= \text{MLP}\left(\mathbf{h}_i, \mathbf{h}_j, \mathbf{h}_i \| \mathbf{h}_j\right), e_{ij} \in \mathcal{E},
\end{aligned}
\tag{1}
$$

here $\|$ denotes the concatenation operator. To ensure the sampling process from $w_{ij}$ is differentiable and facilitate gradient-based optimization, we leverage the Gumbel-Softmax reparameterization trick (Bengio et al., 2013; Maddison et al., 2016), which is applied as follows:

$$
\begin{aligned}
p_{ij} &= \sigma\left((\log \epsilon - \log(1 - \epsilon) + \omega_{ij})/\tau\right), \epsilon \sim \mathcal{U}(0, 1), \\
\widetilde{\mathbf{A}}_{ij} &= 1 - \text{sg}(p_{ij}) + p_{ij},
\end{aligned}
\tag{2}
$$

here $\widetilde{\mathbf{A}}$ denotes the sampled adjacency matrix, $\tau$ is the temperature, $sg(\cdot)$ denotes the stop-gradient operator, and $\mathcal{U}(0, 1)$ denotes the uniform distribution. $\mathbf{A}_{ij}$ is the edge weight for $e_{ij}$, which remains binary and differentiable for the gradient-based optimization.

Next, we introduce the proposed OOD objectives in `PrunE` that directly act on the subgraph selector to prune spurious edges: (1) *Graph size constraint*, which excludes a portion of uninformative spurious edges by limiting the total edge weights in the graph; (2) *$\epsilon$-probability alignment*, which further suppresses the presence of uninformative edges by aligning the predicted probabilities of certain edges close to zero.

**Graph size constraint.** We first introduce a regularization term $\mathcal{L}_e$ which encourages a graph size distinction between $\widetilde{G} \sim t(G)$ and $G$:

$$\mathcal{L}_e = \mathbb{E}_\mathcal{G} \left( \frac{\sum_{(i,j)\in\mathcal{E}} \widetilde{\mathbf{A}}_{ij}}{|\mathcal{E}|} - \eta \right)^2, \qquad (3)$$

where $\eta$ is a hyper-parameter that controls the budget for the total number of edges pruned by $t(\cdot)$. The core insight is that when $\mathcal{L}_e$ acts as a regularization term for ERM, the subgraph selector will prune spurious edges while preserving edges in $G_c$, since ERM learns all useful patterns that are highly correlated with the target labels (Kirichenko et al., 2023). Therefore, given Assumption 2.1, $G_c$ will be preserved due to its strong correlation to the targets, and a subset of edges in $G_s$ will be excluded. Notably, it is critical to select a suitable $\eta$, as an overly aggressive choice, e.g., $\eta = 0.1$, may result in pruning edges in $G_c$ as well. In practice, we find that $\eta \in \{0.75, 0.85\}$ works well for most datasets. In Proposition 5.1, we demonstrate that the graph size regularization $\mathcal{L}_e$ provably prunes spurious edges while retaining invariant edges.

**$\epsilon$-probability alignment.** Although $\mathcal{L}_e$ is able to prune a subset of spurious edges, it is challenging to get rid off all spurious edges. To further suppress the occurence of spurious edges, we propose the following regularization on $t(\cdot)$:

$$\mathcal{L}_s = \mathbb{E}_\mathcal{G} \frac{1}{|\mathcal{E}_s|} \sum_{e_{ij}\in\mathcal{E}_s} |p_{ij} - \epsilon| . \qquad (4)$$

Here, $\epsilon$ is a value close to zero, $p_{ij}$ denotes the normalized probability of the edge $e_{ij}$, and $\mathcal{E}_s$ is the lowest $K\%$ of edges among all estimated edge weights $w_{ij} \in \mathcal{E}$ by the subgraph selector $t(\cdot)$.

The key insight is that edges from $G_c$ are likely to exhibit higher predicted probabilities compared to edges in $G_s$. Thus, by aligning the bottom $K\%$ edges with the lowest predicted probability to a small probability score $\epsilon$, it becomes more likely to suppress spurious edges rather than invariant edges. When $K$ gets larger, $\mathcal{L}_s$ will inevitably push down the probabilities of edges in $G_c$. However, ERM will drive up the probabilities of informative edges for accurate prediction, ensuring that the important edges are included in $\widehat{G}_c$. Therefore, the penalty for $\mathcal{L}_s$ should be relatively small compared to the penalty of ERM. In practice, we find that $\lambda_2 \in \{1e-2, 1e-3\}$ and $K = 90$ work stably across most datasets. In all experiments, we set $\epsilon = \frac{1}{|\mathcal{E}|}$, which works well for all the datasets.

**Final objective.** The overall objective is formulated as:

$$\mathcal{L} = \mathcal{L}_{GT} + \lambda_1 \mathcal{L}_e + \lambda_2 \mathcal{L}_s, \qquad (5)$$

here $\lambda_i, i \in \{1, 2\}$ are hyperparameters that balance the contribution of each component to the overall objective, and

$\mathcal{L}_{GT}$ denotes the ERM objective:

$$\mathcal{L}_{GT} = -\mathbb{E}_\mathcal{G} \sum_{k\in\mathcal{C}} Y_k \log \left( f(t(G))_k \right), \qquad (6)$$

where $Y_k$ denotes the class label $k$ for graph $G$, $f(t(G))_k$ is the predicted probability for class $k$ of graph $G$.

---

**Algorithm 1** The proposed method

1: **Input:** Graph dataset $\mathcal{G}$, epochs $E$, learning rates $\eta$, hyperparameters $\lambda_1, \lambda_2$
2: **Output:** Optimized GNN model $f^* = \rho^* \circ h^*$, and the subgraph selector $t^*(\cdot)$.
3: **Initialize:** GNN encoder $h(\cdot)$, classifier $\rho(\cdot)$, and the learnable data transformation $t(\cdot)$.
4: **for** epoch $e = 1$ **to** $E$ **do**
5:     **for** each minibatch $\mathcal{B} \in \mathcal{G}$ **do**
6:         Calculate $w_{ij}$ using Eqn. 1 for each graph $G \in \mathcal{B}$
7:         Calculate $\mathcal{L}_e$ using Eqn. 3
8:         Calculate $\mathcal{L}_s$ using Eqn. 4
9:         Sample $\widetilde{G} \sim t(G)$ using $t(\cdot)$ for each $G \in \mathcal{B}$
10:        Calculate cross-entropy loss $\mathcal{L}_{GT}$ using Eqn. 6
11:        Compute the total loss $\mathcal{L} = \mathcal{L}_{GT} + \lambda_1 \mathcal{L}_e + \lambda_2 \mathcal{L}_{div}$
12:        Perform backpropagation to update the parameters of $h(\cdot), \rho(\cdot)$, and $t(\cdot)$
13:     **end for**
14: **end for**

---

## 5. Theoretical Analysis

In this section, we provide some theoretical analysis on our proposed method PrunE. All proofs are included in Appendix D.

**Proposition 5.1.** *Under Assumption 2.1, the size constraint loss $\mathcal{L}_e$, when acting as a regularizer for the ERM loss $\mathcal{L}_{GT}$, will prune edges from the spurious subgraph $G_s$, while preserving the invariant subgraph $G_c$ given a suitable $\eta$.*

Prop. 5.1 demonstrates that by enforcing graph size constraint, $\mathcal{L}_e$ will only prune spurious edges, thus making the size of $G_s$ to be smaller. Next we show that $\mathcal{L}_e$ provably improves OOD generalization ability by shrinking $|G_s|$.

**Theorem 5.2.** *Let $l((x_i, x_j, y, G); \theta)$ denote the 0-1 loss function for predicting whether edge $e_{ij}$ presents in graph $G$ using $t(\cdot)$, and*

$$L(\theta; D) := \frac{1}{n} \sum_{(x_i,x_j,y,G)\sim D} l((x_i, x_j, y, G); \theta), \forall e_{ij} \in \mathcal{E}.$$

$$L(\theta; S) := \frac{1}{m} \sum_{(x_i,x_j,y,G)\sim S} l((x_i, x_j, y, G); \theta), \forall e_{ij} \in \mathcal{E}.$$

$$(7)$$

*where $D$ and $S$ represent the training and test set distributions, respectively, $c$ is a constant, and $n$ and $m$ denotes the sample size in training set and test set respectively. Then, with probability at least $1 - \delta$ and $\forall \theta \in \Theta$, we have:*

$$|L(\theta; D) - L(\theta; S)| \le 2(c|G_s| + 1)M, \qquad (8)$$

where $M = \sqrt{\frac{\ln(4|\Theta|) - \ln(\delta)}{2n}} + \sqrt{\frac{\ln(4|\Theta|) - \ln(\delta)}{2m}}$.

Theorem 5.2 establishes an OOD generalization bound that incorporates $|G_s|$ due to domain shifts. When $|G_s| = 0$, Eqn. 8 reduces to the traditional in-distribution generalization bound. Theorem 5.2 shows that $\mathcal{L}_e$ enhances the OOD generalization ability by reducing the size of $G_s$ and tightens the generalization bound.

**Theorem 5.3.** *Let* $\Theta^* = \arg\inf_{\Theta} \mathcal{L}(\Theta)$*, where* $\Theta^* = \{\rho^*(\cdot), h^*(\cdot), t^*(\cdot)\}$*. For any graph G with target label* $y \in \mathcal{Y}$*, we have* $G_c \approx \mathbb{E}_G[t^*(G)]$*. Consequently, sampling from* $t^*(G)$ *in expectation will retain only the invariant subgraph* $G_c$*, which remains stable and sufficiently predictive for the target label* $y$*.*

Theorem 5.3 demonstrates the ability to retain only $G_c$ by sampling from $t^*(G)$. While previous methods aim to directly identify $G_c$, `PrunE` is able to achieve the similar goal more effectively by pruning spurious edges.

## 6. Related Work

**OOD generalization on graphs.** To tackle the OOD generalization challenge on graph, various methods have been proposed recently. MoleOOD (Yang et al., 2022), GIL (Li et al., 2022b) and MILI (Wang et al., 2024) aim to learn graph invariant features with environment inference. CIGA (Chen et al., 2022) adopts supervised contrastive learning to identify invariant subgraphs for OOD generalization. Several methods (Wu et al., 2022b; Liu et al., 2022; Sui et al., 2023; Jia et al., 2024; Li et al., 2024) utilize graph data augmentation to enlarge the training distribution without perturbing the stable patterns in the graph, enabling OOD generalization by identifying stable features across different augmented environments. SizeShiftReg (Buffelli et al., 2022) proposes a method for size generalization for graph-level classification using coarsening techniques. GSAT (Miao et al., 2022) and DGIB (Yuan et al., 2024) use the information bottleneck principle (Tishby & Zaslavsky, 2015) to identify the minimum sufficient subgraph that explains the model's prediction. Many existing methods attempt to directly identify the invariant subgraph to learn invariant features. However, this approach can be error-prone, especially when spurious substructures exhibit strong correlations with the targets, leading to the failure to preserve the invariant substructure and ultimately limiting the OOD generalization capability. In contrast, `PrunE` aims to exclude spurious edges without directly identifying invariant edges, resulting in preserving the invariant substructure more effectively, and enhanced generalization performance.

**Feature learning in the presence of spurious features.** Several studies have explored the inductive bias and SGD training dynamics of neural networks in the presence of spurious features (Pezeshki et al., 2021; Rahaman et al.,

2019; Shah et al., 2020). Shah et al. (2020) shows that in certain scenarios neural networks can suffer from simplicity bias and rely on simple spurious features, while ignoring the core features. More recently, Kirichenko et al. (2023) has found that even when neural networks heavily rely on spurious features, the core (causal) features can still be learned sufficiently well. Inspired by Kirichenko et al. (2023), the subgraph selector should be able to include $G_c$ to encode invariant features using ERM as the learning objective, given that $G_c$ is both strongly correlated with and predictive of the targets (Assumption 2.1). This insight motivates us to propose a pruning-based graph OOD method. Compared to previous approaches, `PrunE` is capable of preserving a more intact set of edges from $G_c$ to enhance OOD performance, at the cost that certain spurious edges may remain difficult to eliminate.

## 7. Experiments

In this section, we evaluate the effectiveness of `PrunE` on both synthetic datasets and real-world datasets, and answer the following research questions. **RQ1.** How does our method perform compared with SOTA baselines? **RQ2.** How do the individual components and hyperparameters in `PrunE` affect the overall performance? **RQ3.** Can the optimal subgraph selector $t^*(G)$ correctly identify $G_c$? **RQ4.** Do edges in $G_c$ predicted by $t(\cdot)$ exhibit higher probability scores than edges in $G_s$? **RQ5.** How does `PrunE` perform on datasets with concept shift? **RQ6.** How do different GNN architectures impact the OOD performance? More details on the datasets, experiment setup and experimental results are presented in Appendix G.

### 7.1. Experimental Setup

**Datasets.** We adopt GOOD datasets (Gui et al., 2022), OGBG-Molbbbp datasets (Hu et al., 2020; Wu et al., 2018), and DrugOOD datasets (Ji et al., 2022) to comprehensively evaluate the OOD generalization performance of our proposed framework.

**Baselines.** Besides ERM (Vapnik, 1995), we compare our method against two lines of OOD baselines: (1) OOD algorithms on Euclidean data, including IRM (Arjovsky et al., 2020), VREx (Krueger et al., 2021), and Group-DRO (Sagawa et al., 2019); (2) graph-specific OOD methods and data augmentation methods, including DIR (Wu et al., 2022b), GSAT (Miao et al., 2022), GREA (Liu et al., 2022), DisC (Fan et al., 2022), CIGA (Chen et al., 2022), AIA (Sui et al., 2023), DropEdge (Rong et al., 2019), $\mathcal{G}$-Mixup (Han et al., 2022), FLAG (Kong et al., 2022), and LiSA (Yu et al., 2023).

**Evaluation.** We report the ROC-AUC score for GOOD-HIV, OGBG-Molbbbp, and DrugOOD datasets, where the tasks

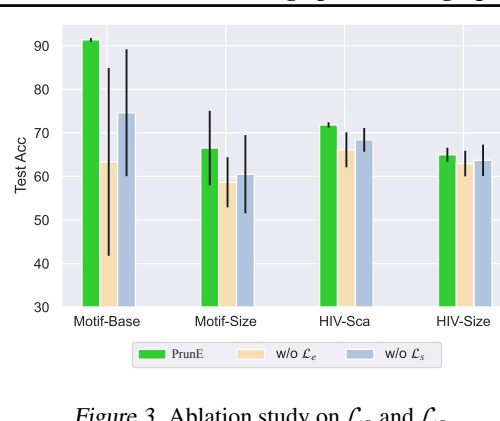

Figure 3. Ablation study on $\mathcal{L}_e$ and $\mathcal{L}_s$.

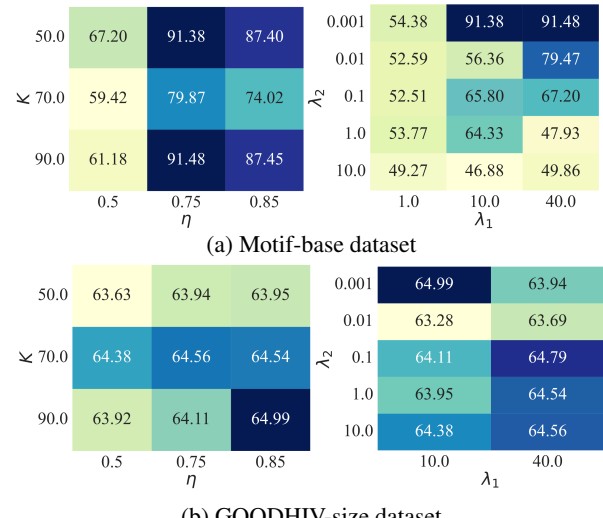

(a) Motif-base dataset

(b) GOODHIV-size dataset

Figure 4. Hyperparameter sensitivity. $\eta \in \{0.75, 0.85\}$ and $K = 90$ yield stable performance across various datasets.

are binary classification. For GOOD-Motif and SPMotif datasets, we use accuracy as the evaluation metric. We run experiments 4 times with different random seeds, select models based on the validation performance, and report the mean and standard deviations on the test set.

## 7.2. Experimental Results

In this section, we report the main results on both synthetic and real-world datasets.

**Synthetic datasets.** The GOOD-Motif datasets fully align with our assumptions, making them a suitable benchmark for evaluating the effectiveness of our proposed method. Our approach outperforms second-best method AIA by 24.19% and 19.13% in Motif-base and Motif-size datasets respectively. This demonstrates the excellent OOD generalization capability of PrunE by pruning spurious edges. While most (graph) OOD methods perform similarly, or even underperform ERM, PrunE outperforms all the baseline methods by a large margin. Notably, the in-distribution performance of ERM on Motif-base dataset is 92.60% (Gui et al., 2022), while our approach achieves a comparable result of 91.48%, further highlighting the superiority of the novel paradigm of pruning spurious edges over the traditional approach of directly identifying invariant edges.

**Real-world datasets.** In real-world datasets, which present more complex and realistic distribution shifts, many graph OOD algorithms exhibit instability, occasionally underperforming ERM. In contrast, our approach consistently achieves stable and superior performance across a diverse set of distribution shifts, and outperform the second-best method by an average of 2.38% in 7 real-world datasets. This also demonstrates that the proposed pruning-based method can be effectively applied to various real-world scenarios, highlighting its applicability.

## 7.3. Ablation Study

In this section, $\eta$ and $K$ we evaluate the impact of $\mathcal{L}_e$ and $\mathcal{L}_{div}$ using the GOODMotif and GOODHIV datasets by

setting $\lambda_1 = 0$ or $\lambda_2 = 0$ in Eqn. 5 to observe the impacts on model performance. As illustrated in Figure 3, removing either $\mathcal{L}_e$ or $\mathcal{L}_s$ leads to a significant drop in test performance across all datasets, and a larger variance. The removal of $\mathcal{L}_e$ results in a more significant decline, as this regularization penalty is stronger (e.g., $\lambda_1$ is set to 10 in the experiments). However, even with $\mathcal{L}_e$, some spurious edges may still exhibit high probabilities, potentially inducing a large variance. By further employing $\mathcal{L}_s$, PrunE effectively reduce predicted probabilities for most spurious edges, thus further reduce the variance and improve the performance.

## 7.4. Hyper-parameter Sensitivity

We study the impact of hyperparameter sensitivity on the edge budget $\eta$ in $\mathcal{L}_e$ and the $K\%$ edges with the lowest probability in $\mathcal{L}_s$. Additionally, we investigate the effects of varying the penalty weights for $\mathcal{L}_e$ and $\mathcal{L}_s$ (i.e., $\lambda_1$ and $\lambda_2$). As illustrated in Figure 4, an unsuitable choice of $\eta$ can negatively impact test performance, e.g., in the GOOD-Motif dataset with *base* split, setting $\eta = 0.5$ may prune too many edges, potentially corrupting $G_c$ and consequently reducing test performance. However, with a suitable $\eta$, test performance remains stable across different values of $K$. Notably, a larger $K$ (e.g., $K = 90$) consistently leads to optimal performance. Readers may raise concerns that a large value of $K$ could also push down the probability of edges in $G_c$, seemingly contradicting the optimal test performance observed at $K = 90$. However, since ERM exerts a stronger influence in driving up the probability of invariant edges to achieve accurate predictions, this compensates for the negative effect of $\mathcal{L}_s$. Notably, when $\lambda_2 \in \{1, 10\}$, the test performance declines significantly for $\forall K \in \{50, 70, 90\}$, as the regularization strengths surpass those of ERM. Re-

*Table 1.* Performance on synthetic and real-world datasets. Numbers in **bold** indicate the best performance, while the underlined numbers indicate the second best performance. ∗ denotes the test performance is statistically significantly better than the second-best method, with $p$-value less than 0.05.

| Method | GOODMotif | | GOODHIV | | EC50 | | | OGBG-Molbbbp | |
|---|---|---|---|---|---|---|---|---|---|
| | base | size | scaffold | size | scaffold | size | assay | scaffold | size |
| ERM | 68.66±4.25 | 51.74±2.88 | 69.58±2.51 | 59.94±2.37 | 62.77±2.14 | 61.03±1.88 | 64.93±6.25 | 68.10±1.68 | 78.29±3.76 |
| IRM | 70.65±4.17 | 51.41±3.78 | 67.97±1.84 | 59.00±2.92 | 63.96±3.21 | 62.47±1.15 | 72.27±3.41 | 67.22±1.15 | 77.56±2.48 |
| GroupDRO | 68.24±8.92 | 51.95±5.86 | 70.64±2.57 | 58.98±2.16 | 64.13±1.81 | 59.06±1.50 | 70.52±3.38 | 66.47±2.39 | 79.27±2.43 |
| VREx | 71.47±6.69 | 52.67±5.54 | 70.77±2.84 | 58.53±2.88 | 64.23±1.76 | 63.54±1.03 | 68.23±3.19 | 68.74±1.03 | 78.76±2.37 |
| DropEdge | 45.08±4.46 | 45.63±4.61 | 70.78±1.38 | 58.53±1.26 | 63.91±2.56 | 61.93±1.41 | 73.79±4.06 | 66.49±1.55 | 78.32±3.44 |
| $G$-Mixup | 59.66±7.03 | 52.81±6.73 | 70.01±2.52 | 59.34±2.43 | 61.90±2.08 | 61.06±1.74 | 69.28±1.36 | 67.44±1.62 | 78.55±4.16 |
| FLAG | 61.12±5.39 | 51.66±4.14 | 68.45±2.30 | 60.59±2.95 | 64.98±0.87 | 64.28±0.54 | 74.91±1.18 | 67.69±2.36 | 79.26±2.26 |
| LiSA | 54.59±4.81 | 53.46±3.41 | 70.38±1.45 | 52.36±3.73 | 62.60±3.62 | 60.96±1.07 | 69.73±0.62 | 68.11±0.52 | 78.62±3.74 |
| DIR | 62.07±8.75 | 52.27±4.56 | 68.07±2.29 | 58.08±2.31 | 63.91±2.92 | 61.91±3.92 | 66.13±3.01 | 66.86±2.25 | 76.40±4.43 |
| DisC | 51.08±3.08 | 50.39±1.15 | 68.07±1.75 | 58.76±0.91 | 59.10±5.69 | 57.64±1.57 | 61.94±7.76 | 67.12±2.11 | 56.59±10.09 |
| CAL | 65.63±4.29 | 51.18±5.60 | 67.37±3.61 | 57.95±2.24 | 65.03±1.12 | 60.92±2.02 | 74.93±5.12 | 68.06±2.60 | 79.50±4.81 |
| GREA | 56.74±9.23 | 54.13±10.02 | 67.79±2.56 | 60.71±2.20 | 64.67±1.43 | 62.17±1.78 | 71.12±1.87 | 69.72±1.66 | 77.34±3.52 |
| GSAT | 60.42±9.32 | 53.20±8.35 | 68.66±1.35 | 58.06±1.98 | 65.12±1.07 | 61.90±2.12 | 74.77±4.31 | 66.78±1.45 | 75.63±3.83 |
| CIGA | 68.71±10.9 | 49.14±8.34 | 69.40±2.39 | 59.55±2.56 | 65.42±1.53 | 64.47±0.73 | 74.94±1.91 | 64.92±2.09 | 65.98±3.31 |
| AIA | 72.91±5.62 | 55.85±7.98 | 71.15±1.81 | 61.64±3.37 | 64.71±0.50 | 63.43±1.35 | 76.01±1.18 | **70.79**±1.53 | 81.03±5.15 |
| PrunE | **91.48**∗±0.40 | **66.53**∗±8.55 | **71.84**∗±0.61 | **64.99**∗±1.63 | **67.56**∗±0.34 | **65.46**∗±0.88 | **78.01**∗±0.42 | 70.32±1.73 | **81.59**±5.35 |

*Table 2.* Test performance with varying $\epsilon$.

| | Motif-base | Motif-size | EC50-sca |
|---|---|---|---|
| $\epsilon = 0.01$ | 91.63±0.73 | 60.38±8.35 | 77.76±1.11 |
| $\epsilon = 0.1$ | 88.14±0.67 | 62.38±10.76 | 76.65±1.92 |
| $\epsilon = 0.3$ | 80.93±4.33 | 50.65±4.95 | 76.07±2.65 |
| $\epsilon = 0.5$ | 74.52±19.89 | 50.28±8.35 | 75.93±1.27 |
| $\epsilon = \frac{1}{|\mathcal{E}|}$ | 91.48±0.40 | 66.53±8.55 | 78.01±0.42 |

garding real-world datasets, such as GOODHIV-size and other datasets in Appendix G, the test OOD performance demonstrates stability across various hyperparameters, underscoring the robustness of `PrunE`.

Furthermore, we investigate the impact of $\epsilon$ in $\mathcal{L}_s$. As shown in Table 2, the optimal performance is observed when $\epsilon$ is a small value close to zero. However, as $\epsilon$ increases, the test performance declines, especially on synthetic datasets. This decline occurs because larger values of $\epsilon$ weaken the suppression effect, potentially leading to adverse effect that hinder generalization. Notably, when $\epsilon = \frac{1}{|\mathcal{E}|}$, the suppression strength is dynamically adjusted for each graph instance, resulting in stable performance across diverse datasets.

## 7.5. In-depth Analysis

**Can $t^*(\cdot)$ identify $G_c$?** To verify whether $t^*(\cdot)$ can indeed identify $G_c$, we conduct experiments using GOOD-Motif datasets with both *base* and *size* splits. These synthetic datasets are suitable for this analysis as they provide ground-truth labels for edges and nodes that are causally related to the targets. First, we collect the target label for each edge, and the predicted probability score from $t^*(\cdot)$ for correctly predicted samples and plot the ROC-AUC curve for both the

validation and test sets for the two datasets. As illustrated in Figure 5(b), the AUC scores for both datasets exhibit high values, demonstrating that $t^*(\cdot)$ accurately identifies $G_c$, which is consistent with the theoretical insights provided in Theorem 5.3. Figure 5(a) illustrates some visualization results using $t^*(\cdot)$, demonstrating that $t^*(\cdot)$ correctly identify invariant edges from $G_c$. More visualization results for the identified edges using $t^*(\cdot)$ are provided in Appendix G.

**Do edges in $G_c$ exhibit a higher probability than edges in $G_s$?** We assess the probability scores and ranking of edges in $G_c$ compared to those in $G_s$ using the GOOD-Motif datasets. Specifically, we plot the average probability and ranking of edges in $G_c$ over the first 40 epochs (excluding the first 10 epochs for ERM pretraining), using the ground-truth edge labels. As shown in Figure 6, for both the Motif-base and Motif-size datasets, the invariant edges in $G_c$ exhibit high probability scores, ranking among the top 50% in both datasets. This demonstrates that the edges from the invariant subgraph generally get higher predicted probability scores compared to spurious edges. However, certain spurious edges may still be overestimated due to their strong correlation with the target labels.

**How does `PrunE` perform on datasets with concept shift?** In the main results, we use covariate shift to evaluate the OOD performance of various methods, where unseen environments arise in validation and test datasets. We also adopt concept shift to evaluate the effectiveness of `PrunE`, where spurious correlation strength varies in training and test sets. As shown in Table 3, `PrunE` also outperforms the SOTA methods significantly. For Motif-base dataset, most of the methods underperform ERM, while `PrunE` achieves 90.28% test accuracy, which is 8.84% higher than ERM.

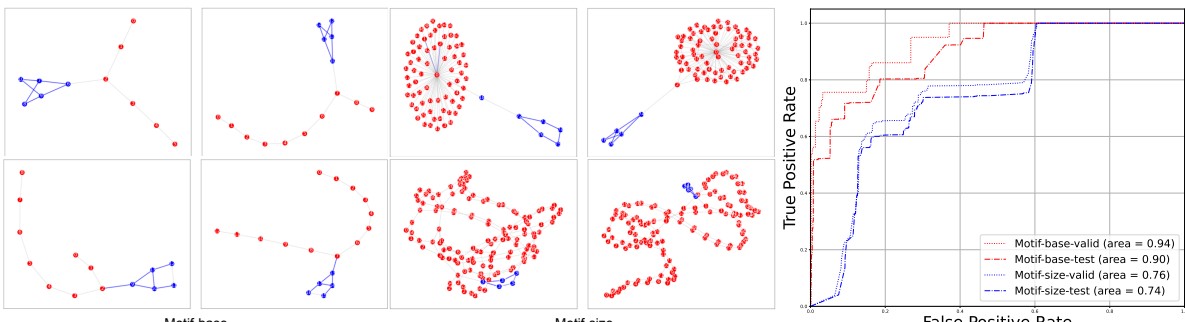

(a) Visualizations on learned subgraph by $t^*(\cdot)$, where blue nodes are ground-truth nodes in $G_c$, and red nodes are ground-truth nodes in $G_s$. The highlighted blue edges are top-K edges predicted by $t^*(\cdot)$, where $K$ is the number of ground-truth invariant edges.

(b) The ROC-AUC curve for predicted edges and ground-truth edges on GOODMotif-base and GOODMotif-size datasets.

*Figure 5.* Empirical visualization and analysis on $t^*(\cdot)$.

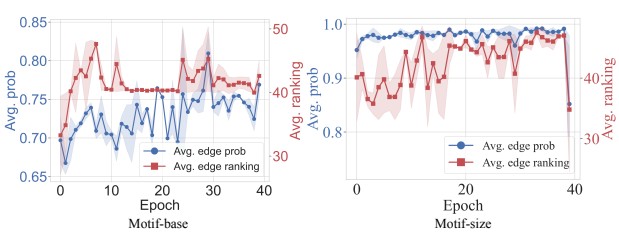

*Figure 6.* Average probability and ranking of edges in $G_c$ for every training epoch. In both datasets, the edges from the invariant subgraph generally get higher predicted probability scores compared to spurious edges. However, certain spurious edges may still be overestimated due to their strong correlation with the target labels.

*Table 3.* Model performance on datasets with concept shift.

| Method | SPMotif | | GOODHIV | GOODMotif |
|--------|---------|---------|---------|-----------|
| | $b = 0.40$ | $b = 0.60$ | size | base |
| ERM | $59.42_{\pm2.63}$ | $60.45_{\pm5.21}$ | $63.26_{\pm2.47}$ | $81.44_{\pm0.45}$ |
| IRM | $59.89_{\pm4.87}$ | $58.10_{\pm4.86}$ | $59.90_{\pm3.15}$ | $80.71_{\pm0.46}$ |
| VRex | $61.16_{\pm3.06}$ | $56.88_{\pm1.19}$ | $60.23_{\pm1.70}$ | $81.56_{\pm0.35}$ |
| GSAT | $64.49_{\pm1.60}$ | $61.27_{\pm1.42}$ | $56.76_{\pm7.16}$ | $76.07_{\pm3.48}$ |
| GREA | $62.08_{\pm4.63}$ | $59.07_{\pm5.94}$ | $60.07_{\pm5.40}$ | $78.27_{\pm4.29}$ |
| CIGA | $65.23_{\pm3.58}$ | $62.17_{\pm2.28}$ | $73.62_{\pm0.86}$ | $81.68_{\pm3.01}$ |
| AIA | $65.11_{\pm2.47}$ | $59.46_{\pm6.23}$ | $74.21_{\pm1.81}$ | $82.51_{\pm2.81}$ |
| PrunE | $\mathbf{67.78_{\pm3.98}}$ | $\mathbf{65.50_{\pm3.53}}$ | $\mathbf{79.50_{\pm1.57}}$ | $\mathbf{90.28_{\pm1.72}}$ |

**How do different GNN encoders affect the model performance?** We examine the effect of using different GNN encoders, specifically GCN (Kipf & Welling, 2017) and GIN (Xu et al., 2018), with the same hidden dimensions and number of layers as $h(\cdot)$. As illustrated in Figure 7, across all four datasets, employing GIN as the feature encoder leads to a increase in test performance. This is likely due to GIN's higher expressivity than GCN (Xu et al., 2018), being as powerful as the 1-WL test (Leman & Weisfeiler, 1968), which allows it to generate more distinguishable features compared to GCN. These enhanced features benefits the optimization of $t(\cdot)$, thereby improving the identification of $G_c$ for OOD generalization. This also highlights another ad-

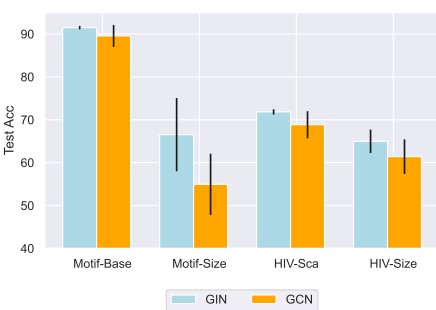

*Figure 7.* Test performance with different GNN encoders. `PrunE` achieves improved OOD performance with more expressive GNN architecture.

vantage of `PrunE`: utilizing a GNN encoder with enhanced expressivity may further facilitate OOD generalization by more accurately identifying $G_c$ through $t(\cdot)$.

## 8. Conclusion

Many graph-specific OOD methods aim to directly identify edges in the invariant subgraph to achieve OOD generalization, which can be challenging and prone to errors. In response, we propose `PrunE`, a pruning-based OOD method that focuses on removing spurious edges by imposing regularization terms on the subgraph selector, without introducing any additional OOD objectives. Through a case study, we demonstrate that, compared to conventional methods, `PrunE` exhibits enhanced OOD generalization capability by retaining more edges in the invariant subgraph. Theoretical analysis and extensive experiments across various datasets validate the effectiveness of this novel learning paradigm. Future research directions include: (1) Extending the pruning-based paradigm to a self-supervised setting without relying on the power of ERM; (2) Expanding this learning paradigm to other scenarios, such as dynamic graphs under distribution shifts.

# Impact Statement

This paper presents work whose goal is to advance the field of Machine Learning. There are many potential societal consequences of our work, none of which we feel must be specifically highlighted here.

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

## A. Notations

We present a set of notations used throughout our paper for clarity. Below are the main notations along with their definitions.

Table 4. Notation Table

| Symbols | Definitions |
|---|---|
| $\mathcal{G}$ | Set of graph datasets |
| $\mathcal{E}_{\text{tr}}$ | Set of environments used for training |
| $\mathcal{E}_{\text{all}}$ | Set of all possible environments |
| $G$ | An undirected graph with node set $\mathcal{V}$ and edge set $\mathcal{E}$ |
| $\mathcal{V}$ | Node set of graph $G$ |
| $\mathcal{E}$ | Edge set of graph $G$ |
| $\mathbf{A}$ | Adjacency matrix of graph $G$ |
| $\mathbf{X}$ | Node feature matrix of graph $G$ |
| $D$ | Feature dimension of node features in $\boldsymbol{X}$ |
| $G_c$ | Invariant subgraph of $G$ |
| $G_s$ | Spurious subgraph of $G$ |
| $\widehat{G}_c$ | Estimated invariant subgraph |
| $\widehat{G}_s$ | Estimated spurious subgraph |
| $|G|$ | The number of edges in graph $G$. |
| $Y$ | Target label variable |
| $\mathbf{w}$ | A vector |
| $\mathbf{W}$ | A matrix |
| $W$ | A random variable |
| $\mathcal{W}$ | A set |
| $f = \rho \circ h$ | A GNN model comprising encoder $h(\cdot)$ and classifier $\rho(\cdot)$ |
| $t(\cdot)$ | Learnable data transformation function for structural modifications |
| $\widetilde{G} \sim t(\cdot)$ | A view sampled from $t(\cdot)$, e.g., $\widetilde{G} \sim t(\cdot)$. We may use $t(G)$ to denote a sampled view from $G$ via $t(\cdot)$, e.g., $I(G; t(G))$ |
| $\mathbf{h}_v$ | Representation of node $v \in \mathcal{V}$ of graph $G$ |

## B. More Preliminaries

**Graph Neural Networks.** In this work, we adopt message-passing GNNs for graph classification due to their expressiveness. Given a simple and undirected graph $G = (\mathbf{A}, \mathbf{X})$ with $n$ nodes and $m$ edges, where $\mathbf{A} \in \{0,1\}^{n \times n}$ is the adjacency matrix, and $\mathbf{X} \in \mathbb{R}^{n \times d}$ is the node feature matrix with $d$ feature dimensions, the graph encoder $h : \mathbb{G} \to \mathbb{R}^h$ aims to learn a meaningful graph-level representation $h_G$, and the classifier $\rho : \mathbb{R}^h \to \mathbb{Y}$ is used to predict the graph label $\widehat{Y}_G = \rho(h_G)$. To obtain the graph representation $h_G$, the representation $\mathbf{h}_v^{(l)}$ of each node $v$ in a graph $G$ is iteratively updated by aggregating information from its neighbors $\mathcal{N}(v)$. For the $l$-th layer, the updated representation is obtained via an AGGREGATE operation followed by an UPDATE operation:

$$\mathbf{m}_v^{(l)} = \text{AGGREGATE}^{(l)} \left( \left\{ \mathbf{h}_u^{(l-1)} : u \in \mathcal{N}(v) \right\} \right), \tag{9}$$

$$\mathbf{h}_v^{(l)} = \text{UPDATE}^{(l)} \left( \mathbf{h}_v^{(l-1)}, \mathbf{m}_v^{(l)} \right), \tag{10}$$

where $\mathbf{h}_v^{(0)} = \mathbf{x}_v$ is the initial node feature of node $v$ in graph $G$. Then GNNs employ a READOUT function to aggregate the final layer node features $\left\{ \mathbf{h}_v^{(L)} : v \in \mathcal{V} \right\}$ into a graph-level representation $\mathbf{h}_G$:

$$\mathbf{h}_G = \text{READOUT} \left( \left\{ \mathbf{h}_v^{(L)} : v \in \mathcal{V} \right\} \right). \tag{11}$$

# C. Additional Related Work

**OOD Generalization on Graphs.** Recently, there has been a growing interest in learning graph-level representations that are robust under distribution shifts, particularly from the perspective of invariant learning. MoleOOD (Yang et al., 2022) and GIL (Li et al., 2022b) propose to infer environmental labels to assist in identifying invariant substructures within graphs. DIR (Wu et al., 2022b), GREA (Liu et al., 2022) and iMoLD (Zhuang et al., 2023) employ environment augmentation techniques to facilitate the learning of invariant graph-level representations. These methods typically rely on the explicit manipulation of unobserved environmental variables to achieve generalization across unseen distributions. AIA (Sui et al., 2023) employs an adversarial augmenter to explore OOD data by generating new environments while maintaining stable feature consistency. To circumvent the need for environmental inference or augmentation, CIGA (Chen et al., 2022) and GALA (Chen et al., 2023) utilizes supervised contrastive learning to identify invariant subgraphs based on the assumption that samples sharing the same label exhibit similar invariant subgraphs. LECI (Gui et al., 2023) and G-Splice (Li et al., 2023b) assume the availability of environment labels, and study environment exploitation strategies for graph OOD generalization. LECI (Gui et al., 2023) proposes to learn a causal subgraph selector by jointly optimizing label and environment causal independence, and G-Splice (Li et al., 2023b) studies graph and feature space extrapolation for environment augmentation, which maintains causal validity. On the other hand, some works do not utilize the invariance principle for graph OOD generalization. DisC (Fan et al., 2022) initially learns a biased graph representation and subsequently focuses on unbiased graphs to discover invariant subgraphs. GSAT (Miao et al., 2022) utilizes information bottleneck principle (Tishby & Zaslavsky, 2015) to learn a minimal sufficient subgraph for GNN explainability, which is shown to be generalizable under distribution shifts. OOD-GNN (Li et al., 2022a) proposes to learn disentangled graph representation by computing global weights of all data.

**Node-level OOD Generalization.** There has been substantial work on OOD generalization for node-level classification tasks. Most existing methods (Wu et al., 2022a; Liu et al., 2023; Li et al., 2023a; Yu et al., 2023) adopt invariant learning to address node-level OOD challenges. Compared to graph-level OOD generalization, node-level OOD problems face unique difficulties, including: (1) distinct types of distribution shifts (e.g., structural or feature-level shifts), (2) non-i.i.d. node dependencies due to the interconnected nature, and (3) computational bottlenecks from subgraph extraction when reducing to graph-level OOD tasks. Due to these challenges, our pruning-based approach cannot be directly extended to node-level tasks. We leave this adaptation to future work.

# D. Proofs of Theoretical Results

## D.1. Proof of Proposition 5.1

*Proof.* We begin by expanding the cross-entropy loss $\mathcal{L}_{GT}$ as:

$$\mathcal{L}_{GT} = -\mathbb{E}_{\mathcal{G}}\left[\log \mathbb{P}(Y \mid f(\widetilde{G}))\right], \tag{12}$$

where $\widetilde{G} \sim t(G)$. Supposing that $|\widetilde{G}| > |G_c|$, which can be controlled by the hyperparameter $\eta$ in Eqn. 3, further assume that $\widetilde{G}$ does not include the invariant subgraph $G_c$. Let a subgraph $g$ be substracted from $\widetilde{G}$ and $|g| = |G_c|$, we then define a new subgraph $G' = \widetilde{G} \setminus g$, and we add $G_c$ to $G'$ to form the new graph $G' \cup G_c$.

Under Assumption 2.1, we know that the invariant subgraph $G_c$ holds sufficient predictive power to $Y$, and $G_c$ is more informative to $Y$ than $G_s$, therefore including $G_c$ will always make the prediction more certain, i.e.,

$$\mathbb{P}(Y \mid f(G' \cup G_c)) > \mathbb{P}(Y \mid f(G' \cup g)), \forall g \subseteq \widetilde{G}, \tag{13}$$

As a result, $\mathcal{L}_{GT}$ will become smaller. Therefore, we conclude that under the graph size regularization imposed by $\mathcal{L}_e$, the optimal solution $\widetilde{G} \sim t(G)$ will always include the invariant subgraph $G_c$, while pruning edges from the spurious subgraph $G_s$. This completes the proof. $\square$

**Remark.** When $\eta$ is set too small, the loss term $\mathcal{L}_e$ may inadvertently prune edges in $G_c$, thereby corrupting the invariant substructure and degrading OOD generalization performance. In practice, we observe that $\eta = \{0.5, 0.75\}$ works well across most datasets stably.

**D.2. Proof of Theorem 5.2**

*Proof.* We first formally define the notations in our proof. Let $l((x_i, x_j, y, G); \theta)$ denote the 0-1 loss for the edge $e_{ij}$ being presented in graph $G$, and

$$L(\theta; D) := \frac{1}{n} \sum_{(x_i, x_j, y, G) \sim D} l((x_i, x_j, y, G); \theta),$$
$$L(\theta; S) := \frac{1}{m} \sum_{(x_i, x_j, y, G) \sim S} l((x_i, x_j, y, G); \theta), \tag{14}$$

where $D$ and $S$ represent the training and test distributions, with $n$ and $m$ being their respective sample sizes. We define:

$$L_c(\theta; D) = \frac{1}{n} \sum_{(x_i, x_j, y, G) \sim D} l((x_i, x_j, y, G_c); \theta), \forall e_{ij} \in G_c.$$
$$L_s(\theta; D) = \frac{1}{n} \sum_{(x_i, x_j, y, G) \sim D} l((x_i, x_j, y, G_s); \theta), \forall e_{ij} \in G_s. \tag{15}$$

Similarly, $L_c(\theta; S)$ and $L_s(\theta; S)$ can be defined for the test distribution. Under Assumption 2.1, $L_c(\theta; D)$ and $L_c(\theta; S)$ are identically distributed due to the stability of $G_c$ across environments, while $L_s(\theta; D)$ and $L_s(\theta; S)$ differ because of domain shifts in $G_s$. We assume:

$$L_s(\theta; \cdot) := c |G_s| L_c(\theta; \cdot), \tag{16}$$

where $c$ is a proportionality constant. As $L_s(\cdot)$ is defined to a summation over all spurious edges, we put $|G_s|$ in the r.h.s to account for this factor. When $|G_s| = 0$, the loss reduces to the in-distribution case $L_c(\theta; \cdot)$.

$$|L(\theta; D) - L(\theta; S)| = |L_c(\theta; D) + L_s(\theta; D) - L_c(\theta; S) - L_s(\theta; S)| \tag{17}$$
$$\leq |L_c(\theta; D) - L_c(\theta; S)| + |L_s(\theta; D) - L_s(\theta; S)| \tag{18}$$
$$= |L_c(\theta; D) - L_c(\theta; S)| + c |G_s| |L_c(\theta; D) - L_c(\theta; S)| \tag{19}$$
$$= (c |G_s| + 1) |L_c(\theta; D) - L_c(\theta; S)|. \tag{20}$$

To bound $|L_c(\theta; D) - L_c(\theta; S)|$, we decompose it as:

$$|L_c(\theta; D) - L_c(\theta; S)| \leq |L_c(\theta; D) - \mathbb{E}[L_c(\theta; D)]| + |\mathbb{E}[L_c(\theta; S)] - L_c(\theta; S)|. \tag{21}$$

Applying Hoeffding's Inequality to each term:

$$\mathbb{P}(|\mathbb{E}[L_c(\theta; D)] - L_c(\theta; D)| \geq \epsilon) \leq 2 \exp(-2\epsilon^2 n), \tag{22}$$
$$\mathbb{P}(|\mathbb{E}[L_c(\theta; S)] - L_c(\theta; S)| \geq \epsilon) \leq 2 \exp(-2\epsilon^2 m). \tag{23}$$

Union bounding over all $\theta \in \Theta$:

$$\mathbb{P}(\exists \theta \in \Theta : |\mathbb{E}[L_c(\theta; D)] - L_c(\theta; D)| \geq \epsilon) \leq 2|\Theta| \exp(-2\epsilon^2 n), \tag{24}$$
$$\mathbb{P}(\exists \theta \in \Theta : |\mathbb{E}[L_c(\theta; S)] - L_c(\theta; S)| \geq \epsilon) \leq 2|\Theta| \exp(-2\epsilon^2 m). \tag{25}$$

Setting both probabilities to $\delta/2$ and solving for $\epsilon$:

$$\epsilon_D = \sqrt{\frac{\ln(4|\Theta|) - \ln(\delta)}{2n}}, \tag{26}$$

$$\epsilon_S = \sqrt{\frac{\ln(4|\Theta|) - \ln(\delta)}{2m}}. \tag{27}$$

Thus, with probability at least $1 - \delta$:

$$|L_c(\theta; D) - L_c(\theta; S)| \le \epsilon_D + \epsilon_S \tag{28}$$

$$= \sqrt{\frac{\ln(4|\Theta|) - \ln(\delta)}{2n}} + \sqrt{\frac{\ln(4|\Theta|) - \ln(\delta)}{2m}}. \tag{29}$$

Substituting into Eqn. 20:

$$|L(\theta; D) - L(\theta; S)| \le 2(c|G_s| + 1) \left( \sqrt{\frac{\ln(4|\Theta|) - \ln(\delta)}{2n}} + \sqrt{\frac{\ln(4|\Theta|) - \ln(\delta)}{2m}} \right). \tag{30}$$

Letting $M = \sqrt{\frac{\ln(4|\Theta|) - \ln(\delta)}{2n}} + \sqrt{\frac{\ln(4|\Theta|) - \ln(\delta)}{2m}}$, we obtain the final bound:

$$|L(\theta; D) - L(\theta; S)| \le 2(c|G_s| + 1)M. \tag{31}$$

$\square$

### D.3. Proof of Theorem 5.3

*Proof.* Our proof consists of the following steps.

**Step 1.** We start by decomposing $\mathbb{E}[t^*(G)]$ into two components: the invariant subgraph $G_c$ and a partially retained spurious subgraph $G_s^{\mathcal{P}}$.

$$\mathbb{E}[t^*(G)] = \mathbb{E}\left[G_c + G_s^{\mathcal{P}}\right]$$
$$= \mathbb{E}\left[G_c\right] + \mathbb{E}\left[G_s^{\mathcal{P}}\right] \tag{32}$$
$$= G_c + \mathbb{E}\left[G_s^{\mathcal{P}}\right]$$

In Eqn. 32, $\mathbb{E}\left[G_c\right] = G_c$ is due to that for any given label $y$, $G_c$ is a constant according to Assumption 2.1, while $G_s^{\mathcal{P}}$ is a random variable.

**Step 2.** We then model $G_s^{\mathcal{P}}$ as a set of independent edges, and calculate the expected total edge weights of $G_c$ and $G_s^{\mathcal{P}}$ respectively. First, we define $W_c$ as the sum of binary random variables corresponding to the edges in $G_c$. Each edge $e_{ij}$ in $G_c$ is associated with a Bernoulli random variable $X_{ij}$ such that:

$$W_c = \sum_{e_{ij} \in G_c} X_{ij}. \tag{33}$$

Similarly, we define $W_s^{\mathcal{P}}$ as the sum of binary random variables corresponding to the edges in $G_s^{\mathcal{P}}$. Each edge $e_{ij}$ in $G_s^{\mathcal{P}}$ is associated with a Bernoulli random variable $X'_{ij}$ such that:

$$W_s^{\mathcal{P}} = \sum_{e_{ij} \in G_s^{\mathcal{P}}} X'_{ij}. \tag{34}$$

$W_c$ and $W_s^{\mathcal{P}}$ are denoted as random r.v. for the total edge weights of $G_c$ and $G_s^{\mathcal{P}}$.

**Step 3.** We then calculate the expected edge weights $\mathbb{E}[W_c]$ and $\mathbb{E}[W_s^{\mathcal{P}}]$ as following.

$$\mathbb{E}[W_c] = \mathbb{E}[\sum_{e_{ij} \in G_c} X_{ij}] = \sum_{e_{ij} \in G_c} \mathbb{E}[X_{ij}] = |G_c|, \tag{35}$$

$$\mathbb{E}[W_s^{\mathcal{P}}] = \mathbb{E}[\sum_{e_{ij} \in G_s^{\mathcal{P}}} X'_{ij}] = \sum_{e_{ij} \in G_s^{\mathcal{P}}} \mathbb{E}[X'_{ij}] = \frac{|G_s^{\mathcal{P}}|}{|\mathcal{E}|} = \frac{\eta|\mathcal{E}| - |G_c|}{|\mathcal{E}|}. \tag{36}$$

Here $\mathcal{E}$ is the set of edges in graph $G$, $\eta|\mathcal{E}|$ is the total edge number limits due to $\mathcal{L}_e$. In Eqn. 35, $\mathbb{E}[X_{ij}] = 1, \forall e_{ij} \in G_c$ is due to that $\mathbb{P}(X_{ij}) = 1$, as $t^*(G)$ always include $G_c$ using the results from Prop. 5.1; In Eqn. 36, $\mathbb{E}[X'_{ij}] = \frac{1}{|\mathcal{E}|}, \forall e_{ij} \in G_s^{\mathcal{P}}$, due to that $\mathbb{P}(X'_{ij}) = \frac{1}{|\mathcal{E}|}$ enforced by $\epsilon$-probability alignment penalty $\mathcal{L}_s$. Therefore, given a suitable $\eta$ that prunes spurious edges from $G_s$, $|\mathcal{E}||G_c| \gg \eta|\mathcal{E}| - |G_c|$, i.e., $\mathbb{E}[t^*(G)]$ will be dominated by $G_c$ in terms of edge probability mass, therefore, we conclude that $G_c \approx \mathbb{E}[t^*(G)]$. □

## E. Complexity Analysis

**Time Complexity.** The time complexity is $\mathcal{O}(CkmF)$, where $k$ is the number of GNN layers, $m$ is the total number of edges in graph $G$, and $F$ is the feature dimensions. Compared to ERM, `PrunE` incurs an additional constant $C > 1$, as it uses a GNN model $t(\cdot)$ for edge selection, and another GNN encoder $h(\cdot)$ for learning feature representations. However, $C$ is a small constant, hence the time cost is on par with standard ERM.

**Space Complexity.** The space complexity for `PrunE` is $\mathcal{O}(C'|\mathcal{B}|mkF)$, where $|\mathcal{B}|$ denotes the batch size. The constant $C' > 1$ is due to the additional subgraph selector $t(\cdot)$. As $C'$ is also a small integer, the space complexity of `PrunE` is also on par with standard ERM.

## F. The Pitfall of Directly Identifying Edges in $G_c$

Most graph-specific OOD methods that model edge probabilities incorporate OOD objectives as regularization terms for ERM. These OOD objectives attempt to directly identify the invariant subgraph for OOD generalization. For example, GSAT (Miao et al., 2022) utilizes the information bottleneck to learn a minimal sufficient subgraph for accurate model prediction; CIGA (Chen et al., 2022) adopt supervised contrastive learning to identify the invariant subgraph that remains stable across different environments within the same class; DIR (Wu et al., 2022b) and AIA (Sui et al., 2023) identify the invariant subgraph through training environments augmentation. However, when spurious substructures exhibit comparable or stronger correlation strength than invariant edges (i.e., edges in $G_c$) with the targets, these methods are unlikely to identify all invariant edges, and preserve the invariant subgraph patterns Since the spurious substructure may be mistakenly identified as the stable pattern. Consequently, while achieving high training accuracy, these methods suffer from poor validation and test performance.

In contrast, `PrunE` avoids this pitfall by proposing OOD objectives that focus on pruning uninformative spurious edges rather than directly identifying causal ones. While strongly correlated spurious edges may still persist, edges in $G_c$ are preserved due to their strong correlation with targets. As a key conclusion, `PrunE` achieves enhanced OOD performance compared to prior methods, as the invariant patterns are more likely to be retained, even if some spurious edges cannot be fully excluded.

## G. More Details about Experiments

### G.1. Datasets details

In our experimental setup, we utilize five datasets: GOOD-HIV, GOOD-Motif, SPMotif, OGBG-Molbbbp, and DrugOOD. The statistics of the datasets are illustrated in Table 5.

**GOOD-HIV** (Gui et al., 2022). GOOD-HIV is a molecular dataset derived from the MoleculeNet (Wu et al., 2018) benchmark, where the primary task is to predict the ability of molecules to inhibit HIV replication. The molecular structures

are represented as graphs, with nodes as atoms and edges as chemical bonds. Following (Gui et al., 2022), We adopt the covariate shift split, which refers to changes in the input distribution between training and testing datasets while maintaining the same conditional distribution of labels given inputs. This setup ensures that the model must generalize to unseen molecular structures that differ in these domain features from those seen during training. We focus on the Bemis-Murcko scaffold (Bemis & Murcko, 1996) and the number of nodes in the molecular graph as two domain features to evaluate our method.

**GOOD-Motif** (Gui et al., 2022). GOOD-Motif is a synthetic dataset designed to test structure shifts. Each graph in this dataset is created by combining a base graph and a motif, with the motif solely determining the label. The base graph type and the size are selected as domain features to introduce covariate shifts. By generating different base graphs such as wheels, trees, or ladders, the dataset challenges the model's ability to generalize to new graph structures not seen during training. We employ the covariate shift split, where these domain features vary between training and testing datasets, reflecting real-world scenarios where underlying graph structures may change.

**SPMotif** (Wu et al., 2022b). In SPMotif datasets (Wu et al., 2022b), each graph comprises a combination of invariant and spurious subgraphs. The spurious subgraphs include three structures (Tree, Ladder, and Wheel), while the invariant subgraphs consist of Cycle, House, and Crane. The task for a model is to determine which one of the three motifs (Cycle, House, and Crane) is present in a graph. A controllable distribution shift can be achieved via a pre-defined parameter $b$. This parameter manipulates the spurious correlation between the spurious subgraph $G_s$ and the ground-truth label $Y$, which depends solely on the invariant subgraph $G_c$. Specifically, given the predefined bias $b$, the probability of a specific motif (e.g., House) and a specific base graph (Tree) will co-occur is $b$ while for the others is $(1 - b)/2$ (e.g., House-Ladder, House-Wheel). When $b = \frac{1}{3}$, the invariant subgraph is equally correlated to the three spurious subgraphs in the dataset.

**OGBG-Molbbbp** (Hu et al., 2020). OGBG-Molbbbp is a real-world molecular dataset included in the Open Graph Benchmark (Hu et al., 2020). This dataset focuses on predicting the blood-brain barrier penetration of molecules, a critical property in drug discovery. The molecular graphs are detailed, with nodes representing atoms and edges representing bonds. Following Sui et al. (2023), we create scaffold shift and graph size shift to evaluate our method. Similarly to Gui et al. (2022), the Bemis-Murcko scaffold (Bemis & Murcko, 1996) and the number of nodes in the molecular graph are used as domain features to create scaffold shift and size shift respectively.

**DrugOOD** (Ji et al., 2022). DrugOOD dataset is designed for OOD challenges in AI-aided drug discovery. This benchmark offers three environment-splitting strategies: Assay, Scaffold, and Size. In our study, we adopt the EC50 measurement. Consequently, this setup results in three distinct datasets, each focusing on a binary classification task for predicting drug-target binding affinity.

*Table 5.* Details about the datasets used in our experiments.

| DATASETS | Split | # TRAINING | # VALIDATION | # TESTING | # CLASSES | METRICS |
|---|---|---|---|---|---|---|
| GOOD-Motif | Base | 18000 | 3000 | 3000 | 3 | ACC |
| | Size | 18000 | 3000 | 3000 | 3 | ACC |
| SPMotif | Correlation | 9000 | 3000 | 3000 | 3 | ACC |
| GOOD-HIV | Scaffold | 24682 | 4113 | 4108 | 2 | ROC-AUC |
| | Size | 26169 | 4112 | 3961 | 2 | ROC-AUC |
| OGBG-Molbbbp | Scaffold | 1631 | 204 | 204 | 2 | ROC-AUC |
| OGBG-Molbace | Scaffold | 1210 | 152 | 151 | 2 | ROC-AUC |
| EC50 | Assay | 4978 | 2761 | 2725 | 2 | ROC-AUC |
| | Scaffold | 2743 | 2723 | 2762 | 2 | ROC-AUC |
| | Size | 5189 | 2495 | 2505 | 2 | ROC-AUC |

### G.2. Detailed experiment setting

**GNN Encoder.** For GOOD-Motif datasets, we utilize a 4-layer GIN (Xu et al., 2018) without Virtual Nodes (Gilmer et al., 2017), with a hidden dimension of 300; For GOOD-HIV datasets, we employ a 4-layer GIN without Virtual Nodes, and with a hidden dimension of 128; For the OGBG-Molbbbp dataset, we adopt a 4-layer GIN with Virtual Nodes, and the dimensions of hidden layers is 64; For the DrugOOD datasets, we use a 4-layer GIN without Virtual Nodes. For SPMotif datasets, we use a 5-layer GIN without Virtual Nodes. All GNN backbones adopt sum pooling for graph readout.

**Training and Validation.** By default, we use Adam optimizer (Kingma & Ba, 2014) with a learning rate of $1e-3$ and a batch size of 64 for all experiments. For DrugOOD, GOOD-Motif and GOOD-HIV datasets, our method is pretrained for 10 epochs with ERM, and for other datasets, we do not use ERM pretraining. We employ an early stopping of 10 epochs according to the validation performance for DrugOOD datasets and GOOD-Motif datasets, and do not employ early stopping for other datasets. Test accuracy or ROC-AUC is obtained according to the best validation performance for all experiments. All experiments are run with 4 different random seeds, the mean and standard deviation are reported using the 4 runs of experiments.

**Baseline setup and hyperparameters.** In our experiments, for the GOOD and OGBG-Molbbbp datasets, the results of ERM, IRM, GroupDRO, and VREx are reported from Gui et al. (2022), while the results for DropEdge, DIR, GSAT, CIGA, GREA, FLAG, $\mathcal{G}$-Mixup and AIA on GOOD and OGBG datasets are reported from Sui et al. (2023). To ensure fairness, we adopt the same GIN backbone architecture as reported in Sui et al. (2023). For the EC50 datasets and SPMotif datasets, we conduct experiments using the provided source codes from the baseline methods. The hyperparameter search is detailed as follows.

For IRM and VREx, the weight of the penalty loss is searched over $\{1e-2, 1e-1, 1, 1e1\}$. For GroupDRO, the step size is searched over $\{1.0, 1e-1, 1e-2\}$. The causal subgraph ratio for DIR is searched across $\{1e-2, 1e-1, 0.2, 0.4, 0.6\}$. For DropEdge, the edge masking ratio is seached over: $\{0.1, 0.2, 0.3\}$. For GREA, the weight of the penalty loss is tuned over $\{1e-2, 1e-1, 1.0\}$, and the causal subgraph size ratio is tuned over $\{0.05, 0.1, 0.2, 0.3, 0.5\}$. For GSAT, the causal graph size ratio is searched over $\{0.3, 0.5, 0.7\}$. For CIGA, the contrastive loss and hinge loss weights are searched over $\{0.5, 1.0, 2.0, 4.0, 8.0\}$. For DisC, we search over $q$ in the GCE loss: $\{0.5, 0.7, 0.9\}$. For LiSA, the loss penalty weights are searched over:$\{1, 1e-1, 1e-2, 1e-3\}$. For $\mathcal{G}$-Mixup, the augmented ratio is tuned over $\{0.15, 0.25, 0.5\}$. For FLAG, the ascending steps are set to 3 as recommended in the paper, and the step size is searched over $\{1e-3, 1e-2, 1e-1\}$. For AIA, the stable feature ratio is searched over $\{0.1, 0.3, 0.5, 0.7, 0.9\}$, and the adversarial penalty weight is searched over $\{0.01, 0.1, 0.2, 0.5, 1.0, 3.0, 5.0\}$.

**Hyperparameter search for `PrunE`.** For `PrunE`, the edge budget $\eta$ in $\mathcal{L}_e$ is searched over: $\{0.5, 0.75, 0.85\}$; $K$ for the $K\%$ edges with lowest probability score in $\mathcal{L}_s$ is searched over:$\{50, 70, 90\}$; $\lambda_1$, $\lambda_2$ for balancing $\mathcal{L}_e$ and $\mathcal{L}_s$ are searched over: $\{10, 40\}$ and $\{1e-1, 1e-2, 1e-3\}$ respectively. The encoder of subgraph selector $t(\cdot)$ is searched over $\{GIN, GCN\}$, with the number of layers: $\{2, 3\}$.

### G.3. More Experimental Results

We provide more experiment details regarding: (1) Experiment results when there are multiple invariant substructures in a graph. (2) Experiment results for more application domains. (3) Ablation study on ERM pretraining. (4) The capability of `PrunE` of identifying spurious edges. (5) More visualization results on GOOD-Motif datasets in Figure 8 and Figure 9. (6) Hyperparameter sensitivity analysis on GOODHIV scaffold, OGBG-Molbbbp, and EC50 assay datasets, in Figure 10.

*Table 6.* Experimental results on SPMotif datasets with 2 invariant subgraphs in each graph.

| Method | SPMotif ($\#G_c = 2$) | | |
| --- | --- | --- | --- |
| | $b = 0.40$ | $b = 0.60$ | $b = 0.90$ |
| ERM | $53.48_{\pm3.31}$ | $52.59_{\pm4.61}$ | $56.76_{\pm8.06}$ |
| IRM | $52.47_{\pm3.63}$ | $55.62_{\pm7.90}$ | $48.66_{\pm2.33}$ |
| VRex | $49.68_{\pm8.66}$ | $48.89_{\pm4.79}$ | $47.97_{\pm2.61}$ |
| GSAT | $59.34_{\pm7.96}$ | $58.43_{\pm10.64}$ | $55.68_{\pm3.18}$ |
| GREA | $64.87_{\pm5.76}$ | $67.66_{\pm6.29}$ | $59.40_{\pm10.26}$ |
| CIGA | $69.74_{\pm6.81}$ | $71.19_{\pm2.46}$ | $65.83_{\pm10.41}$ |
| AIA | $\mathbf{71.61_{\pm2.09}}$ | $72.01_{\pm2.13}$ | $58.14_{\pm4.21}$ |
| PrunE | $70.41_{\pm7.53}$ | $\mathbf{74.61_{\pm3.17}}$ | $\mathbf{66.75_{\pm4.33}}$ |

**Model performance for graphs with multiple invariant subgraphs.** While Assumption 2.1 assumes the existence of a single invariant substructure causally related to each target label, many real-world graph applications (Hu et al., 2020; Gui et al., 2022) may contain multiple such invariant subgraphs. However, Assumption 2.1 can be reformulated to accommodate multiple $G_c$ without compromising the validity of our assumptions and theoretical results. Specifically, suppose there are

$K$ invariant subgraphs, denoted as $G_{c,i}$ for $i \in [K]$. For any specific $G_{c,i}$, the spurious subgraph $G'_s$ can be redefined as $G'_s = G_s \cup \{G_{c,j} \mid j \neq i\}$. Given this redefinition, and under the presence of $G_s$, Assumption 2.1 still holds. Consequently, the assumptions and theoretical results presented in this work remain valid, even when multiple $G_c$ exist within the datasets. To further support our claim, we curated a dataset based on SPMotif (Wu et al., 2022b), where in the train/valid/test datasets, two invariant substructures are attached to the spurious subgraph. Our method performs effectively under this scenario, as shown in Table 6.

The results of ERM, IRM and VRex for GOODHIV-size are obtained from Gui et al. (2022). As shown in the table, our method achieves the best test performance, indicating that `PrunE` effectively handles concept shift by pruning spurious edges.

**Experiment results on more application domains.** To further evaluate the effectiveness of `PrunE` across different application domains, we conduct experiments on GOOD-CMNIST (Gui et al., 2022) and Graph-Twitter (Socher et al., 2013; Yuan et al., 2022) datasets, the evaluation metric for these datasets is accuracy.

*Table 7.* Test performance on GOOD-CMNIST and Graph-Twitter datasets.

| Method | CMNIST | Graph-Twitter |
|--------|--------|---------------|
| ERM | $28.60_{\pm 1.87}$ | $60.47_{\pm 2.24}$ |
| IRM | $27.83_{\pm 2.13}$ | $56.93_{\pm 0.99}$ |
| Vrex | $28.48_{\pm 2.87}$ | $57.54_{\pm 0.93}$ |
| DisC | $24.99_{\pm 1.78}$ | $48.61_{\pm 8.86}$ |
| GSAT | $28.17_{\pm 1.26}$ | $60.96_{\pm 1.18}$ |
| GREA | $29.02_{\pm 3.26}$ | $59.47_{\pm 2.09}$ |
| CIGA | $32.22_{\pm 2.67}$ | $\underline{62.31}_{\pm 1.63}$ |
| AIA | $\mathbf{36.37}_{\pm \mathbf{4.44}}$ | $61.10_{\pm 0.47}$ |
| PrunE | $\underline{33.89}_{\pm 1.65}$ | $\mathbf{63.37}_{\pm \mathbf{0.76}}$ |

As demonstrated in Table 7, `PrunE` also achieves superior performance in application domains beyond molecular applications, indicating its superior OOD performance and broad applicability.

**Ablation study on ERM pretraining.** We conduct ablation study across 5 datasets without using ERM pretraining. The results are presented in Table 8. As illustrated, incorporating ERM pretraining improves OOD performance in most cases, as the GNN encoder is able to learn useful representations before incorporating $\mathcal{L}_e$ and $\mathcal{L}_s$ to train $t(\cdot)$. Intuitively, this facilitates the optimization of $t(\cdot)$, therefore improving the test performance.

*Table 8.* Ablation study on test datasets.

| | Motif-basis | Motif-size | EC50-Assay | EC50-Sca | HIV-size |
|---|---|---|---|---|---|
| w/ pretraining | $91.48_{\pm 0.40}$ | $66.53_{\pm 8.55}$ | $78.01_{\pm 0.42}$ | $67.56_{\pm 1.63}$ | $64.99_{\pm 1.63}$ |
| w/o pretraining | $91.04_{\pm 0.76}$ | $61.48_{\pm 8.29}$ | $76.58_{\pm 2.14}$ | $66.19_{\pm 1.56}$ | $65.46_{\pm 1.85}$ |

**The capability of `PrunE` to identify spurious edges.** To verify the ability of `PrunE` to identify spurious edges while preserving critical edges in $G_c$, we conduct experiments and provide empirical results on *Recall@K* and *Precision@K* on GOODMotif datasets, where $K$ denotes the $K\%$ edges with lowest estimated probability scores. As illustrated in Table 9, `PrunE` is able to identify a subset of spurious edges with precision higher than 90% across all datasets, even with $K = 50$, indicating that `PrunE` can preserve $G_c$, thus enhancing the OOD generalization performance.

**Visualization results on GOOD-Motif datasets.** We provide more visualization results on GOOD-Motif datasets in Figure 8 and Figure 9, in which the blue nodes represent the ground-truth nodes in $G_c$, and blue edges are estimated edges by $t^*(\cdot)$. We visualize top-K edges with highest probability scores derived from $t(\cdot)$. As shown, `PrunE` is able to identify edges in $G_c$, demonstrating the effectiveness of pruning spurious edges, and aligns with the theoretical results from Theorem 5.3.

**Hyperparameter sensitivity.** We provide more experimental results on hyperparameter sensitivity on real-world datasets. As shown in Figure 10, `PrunE` exhibits stable performance across the real-world datasets, highlighting its robustness to

*Table 9.* Recall@K and Precision@K for Motif-base and Motif-size datasets, where $K$ denotes the $K\%$ edges with lowest estimated probability scores.

| K% | Motif-base | | Motif-size | |
|---|---|---|---|---|
| | Recall | Precision | Recall | Precision |
| 10% | 0.1467 | 1.0000 | 0.0963 | 0.9199 |
| 20% | 0.3076 | 0.9831 | 0.2023 | 0.9602 |
| 30% | 0.4556 | 0.9465 | 0.3093 | 0.9735 |
| 40% | 0.6056 | 0.9374 | 0.4153 | 0.9801 |
| 50% | 0.7356 | 0.9017 | 0.5243 | 0.9841 |

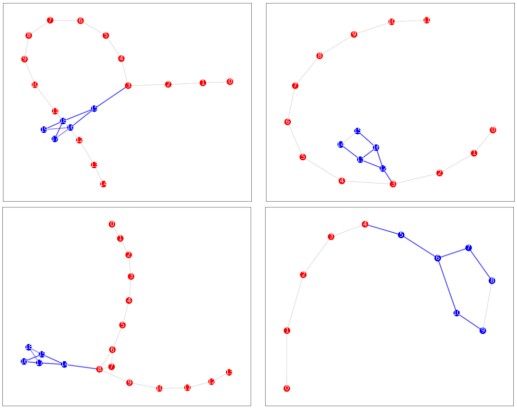 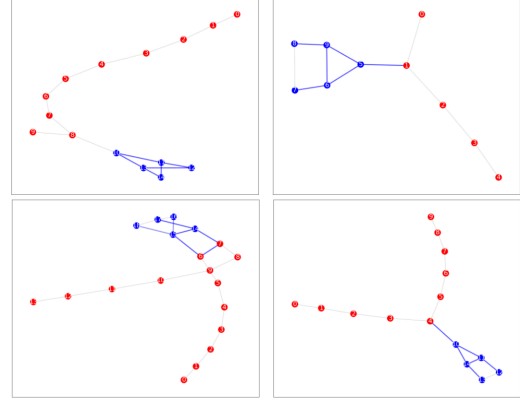

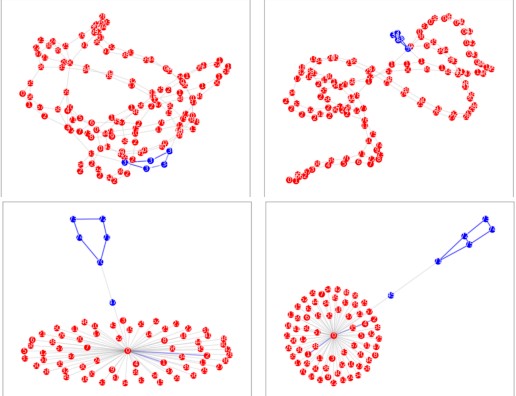 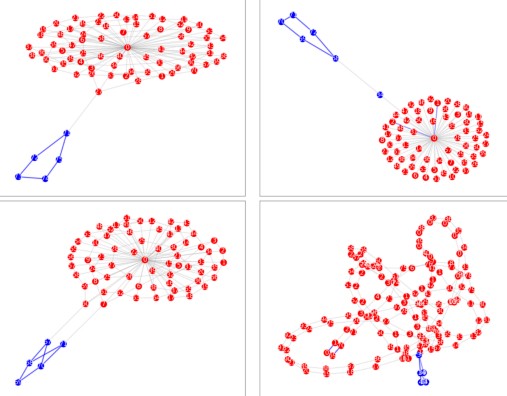

*Figure 8.* More visualization results on Motif-base dataset. The blue nodes are ground-truth nodes in $G_c$, and red nodes are ground-truth nodes in $G_s$. The highlighted blue edges are top-K edges predicted by $t^*(\cdot)$, where $K$ is the number of ground-truth edges from $G_c$ in a graph.

*Figure 9.* More visualization results on Motif-size dataset. The blue nodes are ground-truth nodes in $G_c$, and red nodes are ground-truth nodes in $G_s$. The highlighted blue edges are top-K edges predicted by $t^*(\cdot)$, where $K$ is the number of ground-truth edges from $G_c$ in a graph.

varying hyperparameter configurations.

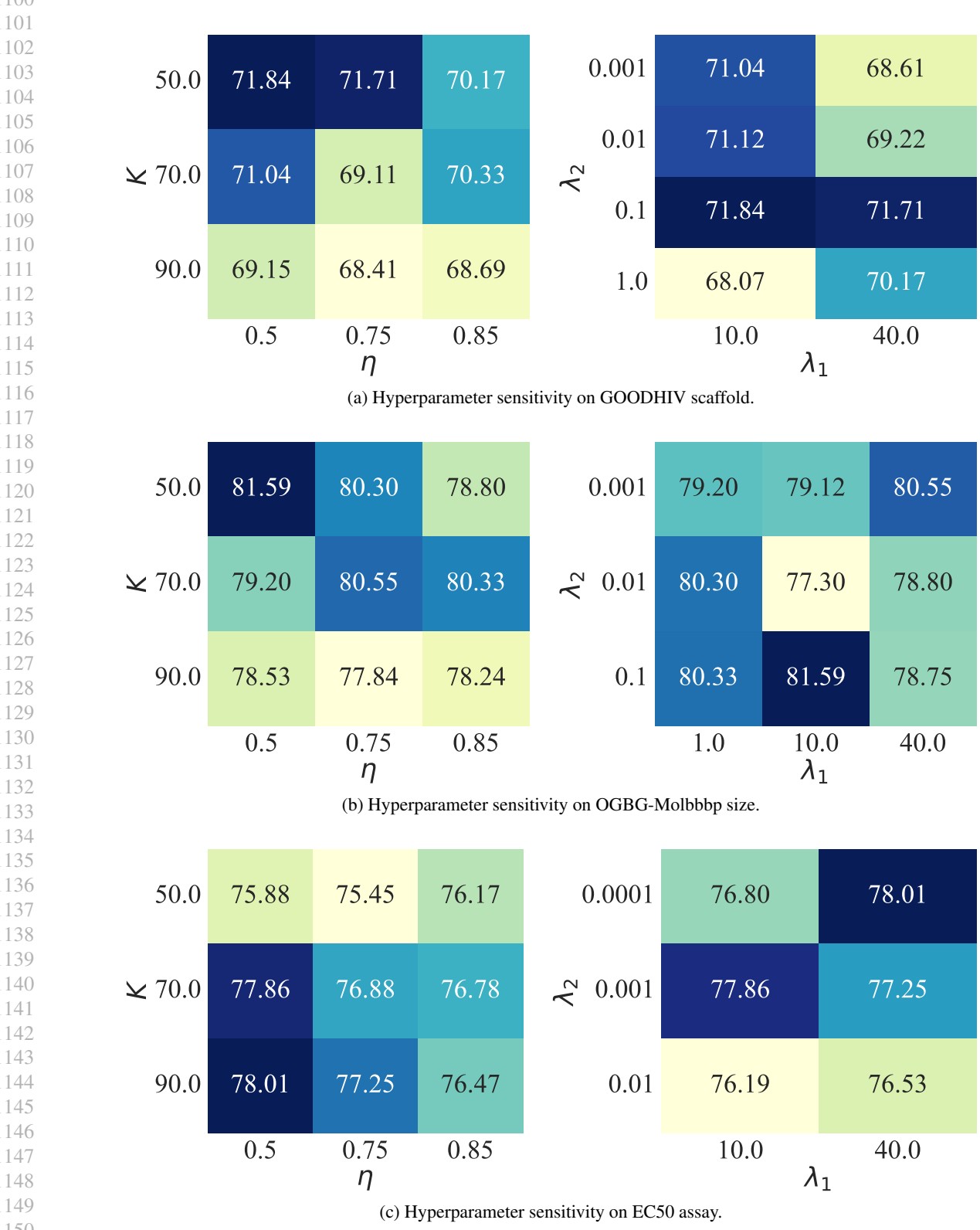

(a) Hyperparameter sensitivity on GOODHIV scaffold.

(b) Hyperparameter sensitivity on OGBG-Molbbbp size.

(c) Hyperparameter sensitivity on EC50 assay.

*Figure 10.* Hyperparameter sensitivity analysis across different datasets.

