# OpenReview forum: "Pruning Spurious Subgraphs for Graph Out-of-Distribtuion Generalization"
_ICML.cc/2025/Conference — Submitted to ICML 2025_

### Official Review · Reviewer_4WXv · 2025-03-12

**Overall Recommendation:** 3

**Summary:**

This paper proposes PrunE, a pruning-based method designed to address the challenge of out-of-distribution (OOD) generalization for Graph Neural Networks (GNNs). Rather than focusing on directly identifying invariant subgraphs, PrunE prunes spurious edges to preserve the invariant subgraph. The method uses two key regularization terms: a graph size constraint to exclude uninformative edges and ϵ-probability alignment to suppress spurious edges. Theoretical analysis and extensive experiments demonstrate that PrunE outperforms existing methods for OOD generalization across multiple benchmark datasets.

**Claims And Evidence:**

Yes, the claims made in the submission are supported by clear and convincing evidence.

**Essential References Not Discussed:**

The paper could cite more works, such as LIRS[1]. This work adopts a similar approach at the representation level, and removing environment representations at the representation level may be more efficient than doing so at the structural level.

[1] Learning Graph Invariance by Harnessing Spuriosity. ICLR 2025

**Experimental Designs Or Analyses:**

Yes, the experimental designs and analysis of the paper are reasonable.

**Methods And Evaluation Criteria:**

Yes

**Other Comments Or Suggestions:**

1.I believe this paper lacks novelty. The motivation of the paper is that directly predicting invariant subgraphs from the graph structure is difficult, so the authors propose using certain techniques to predict the environmental subgraphs and then remove the environmental structure from the graph to identify the invariant subgraphs. However, directly predicting from the structural end seems inefficient, and similar research has already been conducted from the representation end. This makes the paper appear as a product of a feasible but not particularly innovative approach.
2.This paper does not seem to compare its performance with LIRS, a study that shares a similar motivation. Since both papers address similar objectives, I suggest that the authors include LIRS as a baseline method for comparison if possible. Lack of hyperparameter analysis and guidelines for hyperparameter tuning.
3.The paper does not appear to provide an analysis of the time and space complexity of the proposed model.
Similarly, the paper does not provide a comparison of the time and space consumption of the proposed model against other baseline methods. This omission may leave readers uncertain about the efficiency and memory requirements of the model

**Other Strengths And Weaknesses:**

1.The paper presents a novel pruning-based approach to OOD generalization, which is a significant departure from prior methods that focus on direct invariant subgraph identification, leading to better retention of meaningful structural information.
2.Theoretical justifications are well-developed and clearly presented, providing formal guarantees that pruning spurious edges improves OOD generalization.
3.Extensive experiments across multiple datasets demonstrate strong empirical performance, with significant improvements over baseline methods in both synthetic and real-world settings.

**Questions For Authors:**

Regarding the questions, please refer to the weaknesses section in the strengths and weaknesses part.

**Relation To Broader Scientific Literature:**

This work contributes to a deeper understanding of subgraph pruning in graph-based learning tasks, and its success in enhancing OOD generalization could inspire future research on pruning techniques for large-scale, dynamic, and self-supervised graphs. However, the starting point of this paper bears similarities to previous work[1], making its novelty questionable.

[1] Learning Graph Invariance by Harnessing Spuriosity. ICLR 2025

**Theoretical Claims:**

Yes, the proofs for theoretical claims are correct.

---

> ### Author Rebuttal · Authors · 2025-04-01
>
> Thank you for your constructive feedback! Please see below for our responses to your comments and concerns.
>
> > **Q1: Novelty Issue**
>
> __Response:__ One significant distinction between the proposed PrunE method and most existing OOD approaches lies in its learning paradigm. Specifically, PrunE focuses on pruning uninformative spurious edges, rather than directly identifying invariant subgraphs or explicitly learning invariant features, a strategy commonly employed by most prior methods such as IRM, VRex, DIR, GSAT, CIGA, and AIA. Therefore, we respectfully emphasize that PrunE differs from the majority of existing OOD methods in both its learning paradigm and the underlying OOD objectives.
>
> We thank the reviewer for pointing out the related work LIRS which was recently accepted. While PrunE and LIRS share some conceptual commonality, their motivations and methodology differ significantly.
>
> - **Different Motivations.** LIRS aims to learn a more complete set of invariant features through a representation-level approach. In contrast, PrunE is motivated by the difficulty of directly identifying invariant subgraphs. It addresses this by pruning uninformative spurious edges, which facilitates the preservation of invariant substructures.
> - **Technical Design.** LIRS adopts a multi-stage learning paradigm that learning spurious features followed by learning invariant features. In contrast, PrunE employs a single-stage training framework with two novel OOD regularization terms that are distinct from prior work. Compared with LIRS, PrunE presents several unique advantages:
>
>     - **Single-stage training and fewer hyperparameters.** LIRS involves multiple stages, with __nearly 100 hyperparameter combinations__ in total; In contrast, PrunE  demonstrates robust performance with limited set of hyperparameters (as discussed in line 183 and 212) across datasets, greatly reducing model tuning efforts.
>     - **Interpretability.** LIRS operates in latent space and thus lacks interpretability in terms of input structures. PrunE, by operating in the input space, not only being efficient and effective, but also offers interpretability by identifying critical subgraphs that explain the model prediction.
>
> In summary, PrunE is technically and conceptually distinct from LIRS with different motivations, and offers several unique advantages. We appreciate the reviewer’s suggestion and will include a discussion and comparison with LIRS in our revised paper.
>
> > **Q2: Comparison with LIRS and hyperparameter analysis**
>
> __Response:__ Thank you for your thoughtful question. LIRS and PrunE exhibit notable differences in performance across datasets with different features. Specifically, on the *Motif-Base* dataset, PrunE achieves 91.40% accuracy, significantly outperforming LIRS (75.50%). In contrast, on the *Motif-Size* dataset, LIRS performs better than PrunE. This also hightlights the different inductive bias between these two methods. We will add the comparison with LIRS to our revised paper.
>
> Regarding hyperparameter sensitivity, PrunE achieves strong performance using a fixed hyperparameter setting, thereby alleviating the need for hyperparameter search. For further details, please refer to our response to Reviewer `ncwH` due to character limits.
>
> > **Q3: Time and space complexity analysis**
>
> __Response:__ As discussed in Appendix E, the time complexity of PrunE is $\mathcal{O}(CkmF)$, where $k$ is the number of GNN layers, $m$ is #edges, and $F$ is the feature dimension. $C>1$ accounts for the use of both the subgraph selector $t(\cdot)$.
>
> The space complexity is $\mathcal{O}(C'|\mathcal{B}|mkF)$, where $|\mathcal{B}|$ is the batch size and $C'$ reflects the additional memory from $t(\cdot)$. The time and memory cost are both on par with ERM.
>
> To further address the reviewer's concern, we conducted additional experiments to evaluate its runtime and memory consumption as below.
>
> | Memory consumption (in MB) | Motif-base  | Molbbbp |
> |:---:|:---:|:---:|
> |ERM|40.62|32.43|
> |IRM|51.76|36.19|
> |VRex|51.52|35.92|
> |GREA|103.22|76.28|
> |GSAT|90.12|58.02|
> |CIGA|104.43|72.47|
> |AIA|99.29|81.55|
> |LIRS|89.15|107.37|
> |PrunE|74.15|61.07|
> ||
>
> |Running time (in seconds)|Motif-base|Molbbbp|
> |:---:|:---:|:---:|
> |ERM|494.34 ± 117.86|92.42 ± 0.42|
> |IRM|968.94 ± 164.09|151.84 ± 7.53|
> |VRex|819.94 ± 124.54|129.13 ± 12.93|
> |GREA|1612.43 ± 177.36|262.47 ± 45.71|
> |GSAT|1233.68 ± 396.19|142.47 ± 25.71|
> |CIGA|1729.14 ± 355.62|352.14 ± 93.32|
> |AIA|1422.34 ± 69.33|217.36 ± 11.04|
> |LIRS|504.87 ± 24.04|421.32 ± 19.86|
> |PrunE|501.62 ± 7.64|133.35 ± 3.47|
> ||
>
> As PrunE only introduces two lightweight regularization terms on the subgraph selector, it is highly efficient in both runtime and memory consumption (**3.15x faster** than LIRS in Molbbbp), highlighting its advantage in computational efficiency.
>
> ---
>
> We sincerely thank the reviewer for the careful review and insightful feedback. We hope that our responses have adequately addressed your concerns regarding our study.

---

> > ### Comment · Reviewer_4WXv · 2025-04-07
> >
> > I confirm that I have read the author response to my review and will update my review in light of this response as necessary.

---

### Official Review · Reviewer_8qvf · 2025-03-12

**Overall Recommendation:** 4

**Summary:**

In this paper, the authors study the problem of graph-level out-of-distribution (OOD) generalization. Their key claim is learning a more sparse graph structure from the vanilla graph by pruning those spurious edges, which they show is effective in preserving the invariant substructure and thus beneficial for OOD generalization. In implementation, the authors adopt a learnable subgraph selector, which assigns each edge in the graph with a learnable weight. By the proposed loss function, the model is required to make the summation of these weights smaller than the number of edges in the vanilla graph. They further design another loss to align edges with the lowest weights to a small value in order to suppress spurious edges. The authors provide theoretical justification of the proposed method and conduct comprehensive experiments to verify its effectiveness.

***
**Update after Rebuttal**

Thanks the authors for their responses, which have adequately addressed my concerns. Currently, I have no other concerns. I have raised my score to 4.

**Claims And Evidence:**

The claim in this work is clear and reasonable. The authors have provide detailed analysis to demonstrate why pruning spurious edges is helpful for OOD generalization. Intuitively, by assigning spurious edges with smaller weights, the model could focus more on the subgraph structure that is invariant under distribution shift, and thus it could have better OOD generalization ability.

**Essential References Not Discussed:**

From my view, there are no related works that are essential to understanding the key contributions of the paper, but are not currently cited or discussed in the paper.

**Experimental Designs Or Analyses:**

I check the experimental designs and results in the main text. From my view, they are sound and sufficient to support the effectiveness of the proposed method.

**Methods And Evaluation Criteria:**

The experiments are conducted on benchmark datasets, and the experimental setting follows previous studies. The authors also provide some visualization on the learned subgraph selector. From my perspective, these experimental results are sufficient to support the effectiveness of the proposed method.

**Other Comments Or Suggestions:**

Typo: According to Eq. (5), the overall objective is $\mathcal{L} = \mathcal{L} _{GT} + \lambda _1 \mathcal{L} _e + \lambda _2 \mathcal{L} _s$. However, in line 11 of Algorithm 1, the overall objective is $\mathcal{L} = \mathcal{L} _{GT} + \lambda _1 \mathcal{L} _e + \lambda _2 \mathcal{L} _{div}$. I think that the term $\mathcal{L} _{div}$ should be corrected as $\mathcal{L} _s$.

Suggestion: There is not clear definition of the learnable subgraph selector $t(\cdot)$, and I can only know that it is a mapping $t\mathbb{R}^{n \times n} \times \mathbb{R}^{n \times D} \to \mathbb{R}^{n \times n}$. From my understanding, I think that $t(G)$ is defined by resetting each edge $e_ij$ in $\mathcal{E}$ via $e_{ij} \sim \text{Bernoulli}(p_{ij})$. I suggest the authors to clarify this.

**Other Strengths And Weaknesses:**

The strength of this paper is proposing a simple and effective method for graph-level OOD problem. The motivation is clear and reasonable, namely, learning invariant subgraph structure by pruning spurious edges. Also, the experimental results of the proposed method are also impressive. The weakness of this paper is that the proposed method is heuristic, since it simply use the combination of two functions to encourage the model to assign certain edges with small weights and align them to a small value. It is still uncleared whether the model could always correctly find those spurious edges and assign then with small weights. And, the theoretical analysis for the success of this method is also not sufficient. The authors attributes this to ERM, albeit I can not fully agree with them. Since GNN is adopted as the learning model and minimize the loss via stochastic optimization algorithms, the learned parameters are more likely to be a local optima rather than the global optima. From my personal understanding, the model may choose those local optima that spurious are assigned with small weights due to the implicit bias of the learning algorithm. Therefore, analyzing from the perspective of optimization algorithm could be a promising future direction.

**Questions For Authors:**

1. Whether the model could always correctly find those spurious edges and assign then with small weights? If so, why?
2. The proposed framework seems to only apply for the graph-level OOD problem. It is possible to extend it for node-level or edge-level OOD problem?

**Relation To Broader Scientific Literature:**

The key contribution of this work is introducing the idea of learning invariant subgraph structure via pruning spurious edges and designing a simple and effective framework to achieve this. This could bring new insights to the graph learning community, including researchers who focus on OOD problem and others that focus on learning from graphs with noisy structure, namely there exists missing or incorrect edges.

**Theoretical Claims:**

I have carefully checked the proofs in the appendix. The overall proof process is correct. The only place that I am unclear is Eq. (21), where the authors seem to miss the term $\vert \mathbb{E}[L _c (\theta, D)] - \mathbb{E}[L _c (\theta, S)] \vert$. In other words, Eq. (21) holds only when $\mathbb{E}[L _c (\theta, D)] - \mathbb{E}[L _c (\theta, S)]=0$ holds. I encourage the authors to clarify this.

---

> ### Author Rebuttal · Authors · 2025-04-01
>
> Thank you for your thoughtful comments and positive feedback! Please see below for our responses to your comments and concerns.
>
> ---
>
> > **Q1: The effectiveness of PrunE to assign low probabilit y weights to spurious edges**
>
> __Response:__ Thank you for raising this crucial point. Through extensive experiments, we find that the effectiveness of identifying and pruning spurious edges rely on two critical factors:
>
> - The size of the spurious subgraphs $G_s$.
> - The complexity of topological structures of $G_s$.
>
> As $|G_s|$ increases and the spurious structures become more intricate, the performance of all OOD methods tends to degrade. This is primarily because certain spurious substructures may exhibit strong correlations with the target labels, leading to misclassification of invariant substructures and overestimation of spurious edges.
>
> One such example is the *Motif-Base* and *Motif-Size* datasets, where the OOD performance of most methods drops significantly in Motif-size dataset due to the increased size of $G_s$ and the more intricate spurious subgraph topology. Similar to existing methods, PrunE will fail to assign low probabilty score to some spurious edges in these scenarios.
>
> However PrunE is also able to assign high probability scores to invariant edges in $G_c$, while previous methods that attemp to directly identify these edges tend to assign low probability to them. This ability is critical for the improved OOD generalization performance compared to prior approaches that attempt to identify invariant subgraphs directly.
>
> How to further identify and suppress spurious edges that are strongly correlated with target labels remains a challenging problem and represents a promising direction for our future research.
>
> > **Q2: Extending to node-level and edge-level OOD tasks**
>
> __Response:__ Thank you for raising this important point. We have conducted experiments using PrunE on the Cora-Word and Cora-Degree datasets, but the performance is comparable to that of ERM. Similarly, many OOD algorithms, such as IRM, VRex, and GroupDRO, that are effective in the vision domain and graph-level OOD tasks tend to perform on par with or even worse than ERM in node-level OOD settings, as evidenced in [1].
>
> This discrepancy may arise from fundamental differences between the two problem settings. Specifically, in node-level OOD tasks, samples (i.e., nodes) are interconnected and thus not independently and identically distributed, whereas this issue does not arise in vision or graph-level OOD datasets, where each sample is treated independently.
>
> Due to these different characteristics, methods designed for graph-level OOD generalization and those targeting node- or edge-level OOD challenges are typically developed __separately__. In line with PrunE, most existing graph-specific OOD methods, such as DIR, DisC, CAL, GREA, GSAT, CIGA, and AIA, also focus soly on graph-level OOD settings.
>
> > **Q3: A new perspective from optimization**
>
> __Response:__ While our work is inspired by recent findings [2, 3] that ERM tends to learn both invariant and spurious features, we fully agree with the reviewer that analyzing the implicit bias and regularization effects from an optimization perspective in explaining why the learned solution may generalize well is a compelling direction. We thank the reviewer for highlighting this perspective and will consider it in our future research.
>
> > **Q4: Theoretical claims**
>
> __Response:__ Thank you for your careful review! As $L_c(\theta, \cdot)$ is defined as the loss computed on the invariant subgraph which remains unchanged under any distribution shift, it follows that $\mathbb{E}\left[L_c(\theta, D)\right] - \mathbb{E}\left[L\_c(\theta, S)\right] = 0$ under Assumption 1. We have added additional discussion in Appendix D.2 to further clarify this point.
>
> > **Q5: Implementation of subgraph selector $t(\cdot)$**
>
> __Response:__ We appreciate the reviewer’s careful review. The function $t(\cdot)$ is implemented as a GNN model (e.g., a 2-layer GIN) followed by an MLP that models independent edge weights $p_{ij}$, where each edge is treated as a Bernoulli random variable. We have added additional clarification regarding this implementation detail in Section 4 of the revised manuscript.
>
> > **Q6: typos**
>
> __Response:__ Than you for your careful review! We have corrected this typo in the pseudo-codes of Algorithm 1.
>
> ---
>
> We sincerely thank the reviewer for the careful review and insightful feedback. We hope that our responses have adequately addressed your concerns regarding our study.
>
> ---
>
> **References:**
> 1. Gui, et al., GOOD: A Graph Out-of-Distribution Benchmark, NeurIPS 2022
> 2. Kirichenko et al, Last Layer Re-Training is Sufficient for Robustness to Spurious Correlations. ICLR 2023
> 3. Chen et al., Towards Understanding Feature Learning in Out-of-Distribution Generalization. NeurIPS 2023

---

### Official Review · Reviewer_ncwH · 2025-03-14

**Overall Recommendation:** 4

**Summary:**

The authors introduces PrunE, a pruning-based method for enhancing out-of-distribution generalization in GNNs. The method remove spurious edges to address the challenge. Theoretical guarantees is provided and experiments show PrunE obtain better results compared with other methods.

**Claims And Evidence:**

Yes, claims are clear and convincing.

**Essential References Not Discussed:**

To my best of knowledge, no.

**Experimental Designs Or Analyses:**

Yes. The authors have tested the method on datasets with various size and types in different domains.

**Methods And Evaluation Criteria:**

Yes, methods and evaluation make sense.

**Other Comments Or Suggestions:**

N/A

**Other Strengths And Weaknesses:**

Strengths:
-The proposed methods achieve very good results compared with baselines. The experiments are intensive and reasonable.
-The combination of graph size constraint and probability alignment as regularisation terms seem innovative

Weaknesses:
- The performance of PrunE relies on careful tuning of hyperparameters like the graph size constraint and ϵ-probability alignment. The sensitivity analysis in Figure 4 shows that inappropriate choices of η and K can significantly reduce performance. Is there a clear guidance of how to choose these parameters?
- Only test with GCN and GIN. Why not testing on more GNN encoders with potentially better expressiveness?

**Questions For Authors:**

See weaknesses.

**Relation To Broader Scientific Literature:**

The paper builds on existing graph OOD and causal learning literature but introduces a new paradigm of pruning spurious edges rather than directly identifying invariant subgraphs. This approach aligns with recent advances in causal learning, feature selection, and information bottlenecks but is innovative in the context of graph-based OOD generalization.

**Theoretical Claims:**

Yes, the theoretical claims seem valid.

---

> ### Author Rebuttal · Authors · 2025-04-01
>
> Thank you for your positive feedback and careful review! Please see below for our responses to your comments and concerns.
>
> ---
>
> > **Q1: Reproducibility issue**
>
> __Response:__ As the official policy permits only figures and tables in the anonymous link, we have requested approval from the conference to share a link to our code. We are currently awaiting the approval and we have made the code available, along with a  instruction on how to use the code.
>
> > **Q2: Hyperparameter sensitivity**
>
> __Response:__ Thank you for raising this important point. While inappropriate choices of $\eta$ and $K$ can indeed lead to performance degradation, we have found that setting $K=90$, $\lambda_1=10$, $\lambda_2=1e-3$, and $\eta \in \\{0.75, 0.85\\}$ yields consistently stable performance across both synthetic and real-world datasets, as discussed in lines 183 and 212 of our paper. This demonstrates a key advantage of PrunE over most existing graph OOD methods, which typically require extensive hyperparameter tuning. We will include additional discussion on the selection of hyperparameters in the _Hyperparameter Sensitivity_ section of the revised paper.
>
> > **Q3: Testing with more expressive GNNs**
>
> __Response:__ Thank you for this insightful question. We primarily adopted GCN and GIN, two GNN architectures with different levels of expressiveness, for the following reasons:
>
> - **Experimental convention.** Prior work in graph-level OOD generalization commonly adopts GCN and GIN as backbone architectures. Similarly, widely-used graph OOD benchmark datasets, such as GOOD [1], typically follow the same practice.
>
> - **Integration with the PrunE framework.** More expressive GNNs typically involve high-order message passing, however it is non-trivial to incorporate __high-order__ message passing into PrunE, particularly in the computation of $\mathcal{L}\_{GT}$ in Eqn. (6). This loss involves computing $f(t(G))$, where $t(G)$ down-weights spurious edges while preserving invariant ones, followed by a GNN encoder operating on the reweighted graph via first-order message passing. For high-order message passing, which involves aggregating information from __non-adjacent nodes__, it is unclear how to prune or control message flow in a principled way, as the pruning operation in $t(\cdot)$ do not naturally apply to higher-order interactions.
>
> - **Diverse designs of more expressive GNNs.** While many expressive GNNs go beyond first-order message passing, they do so in fundamentally different ways. For instance, PPGN [2] captures pairwise node interactions using outer products, while $K$-hop GNNs [3] aggregate messages over $K$-hop neighborhoods, and subgraph-based GNNs [4] extract a rooted subgraph for each node independently. These technical differences imply that a unified pruning mechanism may not apply, and different designs may require distinct treatments for integration with PrunE.
>
> For these reasons, we adopt GCN and GIN, both of which rely on first-order message passing and can be naturally incorporated into the PrunE framework via edge reweighting. Nevertheless, we fully agree with the reviewer that integrating more expressive GNN architectures into PrunE is a promising direction for our future research.
>
> ---
>
> We sincerely thank the reviewer for the careful review and insightful feedback. We hope that our responses have adequately addressed your concerns regarding our study.
>
> ---
>
> **References:**
> 1. Gui, et al., GOOD: A Graph Out-of-Distribution Benchmark, NeurIPS 2022
> 2. Maron, et al., Provably Powerful Graph Networks, NeurIPS 2019
> 3. Nikolentzos, et al., k-hop Graph Neural Networks, Neural Networks
> 4. Zhang, et al., Nested Graph Neural Networks, NeurIPS 2021

---

### Official Review · Reviewer_V6Vq · 2025-03-18

**Overall Recommendation:** 3

**Summary:**

This paper introduces PrunE, a novel pruning-based method to enhance OOD generalization in GNNs. Unlike previous approaches that attempt to directly identify invariant subgraphs, PrunE focuses on pruning spurious edges, preserving the invariant subgraph more effectively. The method employs graph size constraints and ϵ-probability alignment to eliminate spurious edges. The authors provide theoretical guarantees and extensive empirical evaluations, demonstrating that PrunE outperforms existing methods.

**Claims And Evidence:**

Yes

**Essential References Not Discussed:**

I think the current discussion on related work is proper, but I am not quite familiar with the field.

**Experimental Designs Or Analyses:**

It would be beneficial to include an analysis of when and why PrunE fails

**Methods And Evaluation Criteria:**

Yes

**Other Comments Or Suggestions:**

NA

**Other Strengths And Weaknesses:**

Strength: This paper introduces a novel paradigm focusing on removing spurious edges rather than directly identifying edges in $G_c$. By pruning spurious edges, PrunE preserves more edges in $G_c$ than previous methods, thereby improving its OOD generalization performance. The effectiveness of the proposed approach is validated via both theoretical and empirical analyses.

Weakness: The method involves additional regularization terms and subgraph selection, which may introduce computational overhead. A scalability analysis on large-scale datasets should be provided.

**Questions For Authors:**

NA

**Relation To Broader Scientific Literature:**

Yes

**Theoretical Claims:**

I didn't check the details of the proof

---

> ### Author Rebuttal · Authors · 2025-04-01
>
> Thank you for your positive feedback and insightful comments! Please see below for our responses to your comments and concerns.
>
> ---
> > **Q1: When and why PrunE may fail**
>
> __Response:__ Based on our empirical observations, the OOD generalization performance of PrunE, as well as other OOD methods, can be significantly influenced by: __i)__ the size of the spurious subgraphs $G_s$ and __ii)__ the complexity of their topological structures.
>
> As $|G_s|$ increases and the spurious substructures become more intricate, the performance of all methods tends to degrade. This is primarily because certain spurious substructures may exhibit strong correlations with the target labels, leading to misclassification of invariant substructures and overestimation of spurious edges.
>
> One such example is the *Motif-Base* and *Motif-Size* datasets, where the OOD performance of most methods drops significantly in Motif-size dataset due to the increased size of $G_s$ and the more intricate spurious subgraph topology.
>
> While this phenomenon also affects the performance of PrunE, our experimental analysis reveals that, despite occasionally assigning high probabilities to spurious edges (as seen in other methods), PrunE is also able to consistently estimate invariant edges with high confidence. This ability is critical for its superior OOD generalization performance relative to prior approaches.
>
> Nonetheless, how to identify and suppress spurious edges that are strongly correlated with target labels remains a challenging problem and represents a promising direction for our future research.
>
>
> > **Q2: Computational efficiency and scalability**
>
> __Response:__  We thank the reviewer for raising this point. In the context of graph-level OOD generalization, each sample corresponds to an individual graph, typically containing at most a few hundred nodes. This contrasts with node classification tasks, where each sample is a node within a potentially massive graph comprising millions of nodes. As such, scalability is generally not a major concern for graph-level classification datasets.
>
> To further address the reviewer’s concern regarding computational efficiency, we conducted additional experiments to evaluate runtime and memory overhead on two datasets.
>
> __Table 1: Memory consumption of vairous methods (in MB)__
> |  | Motif-base  | Molbbbp |
> |:---:|:---:|:---:|
> | ERM | 40.62 | 32.43 |
> | IRM | 51.76 | 36.19 |
> | VRex | 51.52 | 35.92 |
> | GREA | 103.22 | 76.28 |
> | GSAT | 90.12 | 58.02 |
> | CIGA | 104.43 | 72.47 |
> | AIA | 99.29 | 81.55 |
> | PrunE | 74.15 | 61.07 |
> ||
>
> __Table 2: Runtime of various methods (in seconds)__
> | Method | Motif-base | Molbbbp |
> |:---:|:---:|:---:|
> | ERM | 494.34±117.86 | 92.42±0.42 |
> | IRM | 968.94±164.09 | 151.84±7.53 |
> | VRex | 819.94±124.54 | 129.13±12.93 |
> | GREA | 1612.43±177.36 | 262.47±45.71 |
> | GSAT | 1233.68±396.19 | 142.47±25.71 |
> | CIGA | 1729.14±355.62 | 352.14±93.32 |
> | AIA | 1422.34±69.33 | 217.36±11.04 |
> | PrunE | 501.62±7.64 | 133.35±3.47 |
> ||
>
> As shown in the tables above, compared to most graph-specific OOD methods, PrunE exhibits  advantages in computational efficiency, as it introduces only two lightweight regularization terms on the subgraph selector. In contrast, many existing methods rely on more expensive operations such as data augmentation or contrastive learning. This highlights the computational efficiency of our approach.
>
> ---
>
> We sincerely thank the reviewer for the careful review and insightful feedback. We hope that our responses have adequately addressed your concerns regarding our study.

---

### Decision · Program_Chairs · 2025-05-01

**Decision:**

Reject

**Comment:**

This paper focuses on the out-of-distribution generalization issue in graph neural networks. Instead of directly identifying the causal edges, this paper preserves  the invariant subgraph by pruning spurious edges. Moreover, this paper employs two regularizations, graph size constraint and $\epsilon$-probability alignment, to prune spurious edges. Experimental results demonstrate the effectiveness of the proposed method.


After rebuttal, three reviewers give 'Weak accept' and one reviewer gives 'Accept'.  Reviewer V6Vq holds that this method involves additional regularization terms and subgraph selection, and a scalability analysis on large-scale datasets should be provided. Reviewer 8qvf concerns that the proposed method is heuristic as it  simply uses the combination of two functions to  assign certain edges. The theoretical analysis is also insufficient.  This paper is at the border and given the number of strong submission this year, I unfortunately recommend rejection.